# Bayesian Manifold Learning:
# The Locally Linear Latent Variable Model

**Mijung Park,   Wittawat Jitkrittum,   Ahmad Qamar**,[*]
**Zoltán Szabó,   Lars Buesing**,[†]  **Maneesh Sahani**

Gatsby Computational Neuroscience Unit
University College London

{mijung, wittawat, zoltan.szabo}@gatsby.ucl.ac.uk
atqamar@gmail.com, lbuesing@google.com, maneesh@gatsby.ucl.ac.uk

## Abstract

We introduce the *Locally Linear Latent Variable Model* (LL-LVM), a probabilistic model for non-linear manifold discovery that describes a joint distribution over observations, their manifold coordinates and locally linear maps conditioned on a set of neighbourhood relationships. The model allows straightforward variational optimisation of the posterior distribution on coordinates and locally linear maps from the latent space to the observation space given the data. Thus, the LL-LVM encapsulates the local-geometry preserving intuitions that underlie non-probabilistic methods such as locally linear embedding (LLE). Its probabilistic semantics make it easy to evaluate the quality of hypothesised neighbourhood relationships, select the intrinsic dimensionality of the manifold, construct out-of-sample extensions and to combine the manifold model with additional probabilistic models that capture the structure of coordinates within the manifold.

## 1   Introduction

Many high-dimensional datasets comprise points derived from a smooth, lower-dimensional manifold embedded within the high-dimensional space of measurements and possibly corrupted by noise. For instance, biological or medical imaging data might reflect the interplay of a small number of latent processes that all affect measurements non-linearly. Linear multivariate analyses such as principal component analysis (PCA) or multidimensional scaling (MDS) have long been used to estimate such underlying processes, but cannot always reveal low-dimensional structure when the mapping is non-linear (or, equivalently, the manifold is curved). Thus, there has been substantial recent interest in algorithms to identify non-linear manifolds in data.

Many more-or-less heuristic methods for non-linear manifold discovery are based on the idea of preserving the geometric properties of local neighbourhoods within the data, while embedding, unfolding or otherwise transforming the data to occupy fewer dimensions. Thus, algorithms such as locally-linear embedding (LLE) and Laplacian eigenmap attempt to preserve local linear relationships or to minimise the distortion of local derivatives [1, 2]. Others, like Isometric feature mapping (Isomap) or maximum variance unfolding (MVU) preserve local distances, estimating global manifold properties by continuation across neighbourhoods before embedding to lower dimensions by classical methods such as PCA or MDS [3]. While generally hewing to this same intuitive path, the range of available algorithms has grown very substantially in recent years [4, 5].

---

[*]Current affiliation: Thread Genius
[†]Current affiliation: Google DeepMind

However, these approaches do not define distributions over the data or over the manifold properties. Thus, they provide no measures of uncertainty on manifold structure or on the low-dimensional locations of the embedded points; they cannot be combined with a structured probabilistic model within the manifold to define a full likelihood relative to the high-dimensional observations; and they provide only heuristic methods to evaluate the manifold dimensionality. As others have pointed out, they also make it difficult to extend the manifold definition to out-of-sample points in a principled way [6].

An established alternative is to construct an explicit probabilistic model of the functional relationship between low-dimensional manifold coordinates and each measured dimension of the data, assuming that the functions instantiate draws from Gaussian-process priors. The original *Gaussian process latent variable model* (GP-LVM) required optimisation of the low-dimensional coordinates, and thus still did not provide uncertainties on these locations or allow evaluation of the likelihood of a model over them [7]; however a recent extension exploits an auxiliary variable approach to optimise a more general variational bound, thus retaining approximate probabilistic semantics within the latent space [8]. The stochastic process model for the mapping functions also makes it straightforward to estimate the function at previously unobserved points, thus generalising out-of-sample with ease. However, the GP-LVM gives up on the intuitive preservation of local neighbourhood properties that underpin the non-probabilistic methods reviewed above. Instead, the expected smoothness or other structure of the manifold must be defined by the Gaussian process covariance function, chosen a priori.

Here, we introduce a new probabilistic model over high-dimensional observations, low-dimensional embedded locations and locally-linear mappings between high and low-dimensional linear maps within each neighbourhood, such that each group of variables is Gaussian distributed given the other two. This *locally linear latent variable model* (LL-LVM) thus respects the same intuitions as the common non-probabilistic manifold discovery algorithms, while still defining a full-fledged probabilistic model. Indeed, variational inference in this model follows more directly and with fewer separate bounding operations than the sparse auxiliary-variable approach used with the GP-LVM. Thus, uncertainty in the low-dimensional coordinates and in the manifold shape (defined by the local maps) is captured naturally. A lower bound on the marginal likelihood of the model makes it possible to select between different latent dimensionalities and, perhaps most crucially, between different definitions of neighbourhood, thus addressing an important unsolved issue with neighbourhood-defined algorithms. Unlike existing probabilistic frameworks with locally linear models such as mixtures of factor analysers (MFA)-based and local tangent space analysis (LTSA)-based methods [9, 10, 11], LL-LVM does not require an additional step to obtain the globally consistent alignment of low-dimensional local coordinates.[1]

This paper is organised as follows. In section 2, we introduce our generative model, LL-LVM, for which we derive the variational inference method in section 3. We briefly describe out-of-sample extension for LL-LVM and mathematically describe the dissimilarity between LL-LVM and GP-LVM at the end of section 3. In section 4, we demonstrate the approach on several real world problems.

**Notation:** In the following, a diagonal matrix with entries taken from the vector $\mathbf{v}$ is written $\mathrm{diag}(\mathbf{v})$. The vector of $n$ ones is $\mathbf{1}_n$ and the $n \times n$ identity matrix is $\mathbf{I}_n$. The Euclidean norm of a vector is $\|\mathbf{v}\|$, the Frobenius norm of a matrix is $\|\mathbf{M}\|_F$. The Kronecker delta is denoted by $\delta_{ij}$ ($= 1$ if $i = j$, and 0 otherwise). The Kronecker product of matrices $\mathbf{M}$ and $\mathbf{N}$ is $\mathbf{M} \otimes \mathbf{N}$. For a random vector $\mathbf{w}$, we denote the normalisation constant in its probability density function by $Z_{\mathbf{w}}$. The expectation of a random vector $\mathbf{w}$ with respect to a density $q$ is $\langle \mathbf{w} \rangle_q$.

## 2 The model: LL-LVM

Suppose we have $n$ data points $\{\mathbf{y}_1, \ldots, \mathbf{y}_n\} \subset \mathbb{R}^{d_y}$, and a graph $\mathcal{G}$ on nodes $\{1 \ldots n\}$ with edge set $\mathcal{E}_{\mathcal{G}} = \{(i, j) \mid \mathbf{y}_i \text{ and } \mathbf{y}_j \text{ are neighbours}\}$. We assume that there is a low-dimensional (latent) representation of the high-dimensional data, with coordinates $\{\mathbf{x}_1, \ldots, \mathbf{x}_n\} \subset \mathbb{R}^{d_x}, d_x < d_y$. It will be helpful to concatenate the vectors to form $\mathbf{y} = [\mathbf{y}_1^\top, \ldots, \mathbf{y}_n^\top]^\top$ and $\mathbf{x} = [\mathbf{x}_1^\top, \ldots, \mathbf{x}_n^\top]^\top$.

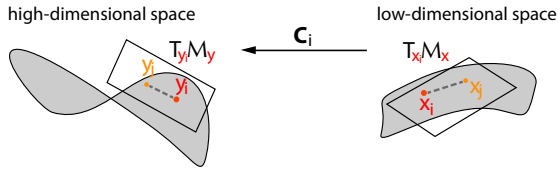

Figure 1: Locally linear mapping $\mathbf{C}_i$ for $i$th data point transforms the tangent space, $T_{\mathbf{x}_i}\mathcal{M}_{\mathbf{x}}$ at $\mathbf{x}_i$ in the low-dimensional space to the tangent space, $T_{\mathbf{y}_i}\mathcal{M}_{\mathbf{y}}$ at the corresponding data point $\mathbf{y}_i$ in the high-dimensional space. A neighbouring data point is denoted by $\mathbf{y}_j$ and the corresponding latent variable by $\mathbf{x}_j$.

Our key assumption is that the mapping between high-dimensional data and low-dimensional coordinates is *locally linear* (Fig. 1). The tangent spaces are approximated by $\{\mathbf{y}_j - \mathbf{y}_i\}_{(i,j)\in\mathcal{E}_{\mathcal{G}}}$ and $\{\mathbf{x}_j - \mathbf{x}_i\}_{(i,j)\in\mathcal{E}_{\mathcal{G}}}$, the pairwise differences between the $i$th point and neighbouring points $j$. The matrix $\mathbf{C}_i \in \mathbb{R}^{d_y \times d_x}$ at the $i$th point linearly maps those tangent spaces as

$$\mathbf{y}_j - \mathbf{y}_i \approx \mathbf{C}_i(\mathbf{x}_j - \mathbf{x}_i). \tag{1}$$

Under this assumption, we aim to find the distribution over the linear maps $\mathbf{C} = [\mathbf{C}_1, \cdots, \mathbf{C}_n] \in \mathbb{R}^{d_y \times n d_x}$ and the latent variables $\mathbf{x}$ that best describe the data likelihood given the graph $\mathcal{G}$:

$$\log p(\mathbf{y}|\mathcal{G}) = \log \iint p(\mathbf{y}, \mathbf{C}, \mathbf{x}|\mathcal{G}) \, \mathrm{d}\mathbf{x} \, \mathrm{d}\mathbf{C}. \tag{2}$$

The joint distribution can be written in terms of priors on $\mathbf{C}, \mathbf{x}$ and the likelihood of $\mathbf{y}$ as

$$p(\mathbf{y}, \mathbf{C}, \mathbf{x}|\mathcal{G}) = p(\mathbf{y}|\mathbf{C}, \mathbf{x}, \mathcal{G})p(\mathbf{C}|\mathcal{G})p(\mathbf{x}|\mathcal{G}). \tag{3}$$

In the following, we highlight the essential components the *Locally Linear Latent Variable Model* (LL-LVM). Detailed derivations are given in the Appendix.

**Adjacency matrix and Laplacian matrix**    The edge set of $\mathcal{G}$ for $n$ data points specifies a $n \times n$ symmetric adjacency matrix $\mathbf{G}$. We write $\eta_{ij}$ for the $i,j$th element of $\mathbf{G}$, which is 1 if $\mathbf{y}_j$ and $\mathbf{y}_i$ are neighbours and 0 if not (including on the diagonal). The graph Laplacian matrix is then $\mathbf{L} = \mathrm{diag}(\mathbf{G}\,\mathbf{1}_n) - \mathbf{G}$.

**Prior on x**    We assume that the latent variables are zero-centered with a bounded expected scale, and that latent variables corresponding to neighbouring high-dimensional points are close (in Euclidean distance). Formally, the log prior on the coordinates is then

$$\log p(\{\mathbf{x}_1 \ldots \mathbf{x}_n\}|\mathbf{G}, \alpha) = -\tfrac{1}{2}\sum_{i=1}^{n}(\alpha\|\mathbf{x}_i\|^2 + \sum_{j=1}^{n}\eta_{ij}\|\mathbf{x}_i - \mathbf{x}_j\|^2) - \log Z_{\mathbf{x}},$$

where the parameter $\alpha$ controls the expected scale ($\alpha > 0$). This prior can be written as multivariate normal distribution on the concatenated $\mathbf{x}$:

$$p(\mathbf{x}|\mathbf{G}, \alpha) = \mathcal{N}(\mathbf{0}, \mathbf{\Pi}), \quad \text{where } \mathbf{\Omega}^{-1} = 2\mathbf{L} \otimes \mathbf{I}_{d_x}, \; \mathbf{\Pi}^{-1} = \alpha\mathbf{I}_{nd_x} + \mathbf{\Omega}^{-1}.$$

**Prior on C**    We assume that the linear maps corresponding to neighbouring points are similar in terms of Frobenius norm (thus favouring a smooth manifold of low curvature). This gives

$$\log p(\{\mathbf{C}_1 \ldots \mathbf{C}_n\}|\mathbf{G}) = -\frac{\epsilon}{2}\Big\|\sum_{i=1}^{n}\mathbf{C}_i\Big\|_F^2 - \frac{1}{2}\sum_{i=1}^{n}\sum_{j=1}^{n}\eta_{ij}\|\mathbf{C}_i - \mathbf{C}_j\|_F^2 - \log Z_{\mathbf{c}}$$

$$= -\frac{1}{2}\mathrm{Tr}\left[(\epsilon\mathbf{J}\mathbf{J}^\top + \mathbf{\Omega}^{-1})\mathbf{C}^\top\mathbf{C}\right] - \log Z_{\mathbf{c}}, \tag{4}$$

where $\mathbf{J} := \mathbf{1}_n \otimes \mathbf{I}_{d_x}$. The second line corresponds to the matrix normal density, giving $p(\mathbf{C}|\mathbf{G}) = \mathcal{MN}(\mathbf{C}|\mathbf{0}, \mathbf{I}_{d_y}, (\epsilon\mathbf{J}\mathbf{J}^\top + \mathbf{\Omega}^{-1})^{-1})$ as the prior on $\mathbf{C}$. In our implementation, we fix $\epsilon$ to a small value[2], since the magnitude of the product $\mathbf{C}_i(\mathbf{x}_i - \mathbf{x}_j)$ is determined by optimising the hyperparameter $\alpha$ above.

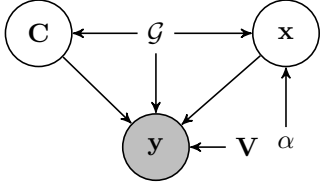

Figure 2: Graphical representation of generative process in LL-LVM. Given a dataset, we construct a neighbourhood graph $\mathcal{G}$. The distribution over the latent variable $\mathbf{x}$ is controlled by the graph $\mathcal{G}$ as well as the parameter $\alpha$. The distribution over the linear map $\mathbf{C}$ is also governed by the graph $\mathcal{G}$. The latent variable $\mathbf{x}$ and the linear map $\mathbf{C}$ together determine the data likelihood.

**Likelihood**  Under the local-linearity assumption, we penalise the approximation error of Eq. (1), which yields the log likelihood

$$\log p(\mathbf{y}|\mathbf{C}, \mathbf{x}, \mathbf{V}, \mathbf{G}) = -\frac{\epsilon}{2}\|\sum_{i=1}^n \mathbf{y}_i\|^2 - \frac{1}{2}\sum_{i=1}^n \sum_{j=1}^n \eta_{ij}(\Delta_{\mathbf{y}_{j,i}} - \mathbf{C}_i \Delta_{\mathbf{x}_{j,i}})^\top \mathbf{V}^{-1}(\Delta_{\mathbf{y}_{j,i}} - \mathbf{C}_i \Delta_{\mathbf{x}_{j,i}}) - \log Z_{\mathbf{y}},$$

(5)

where $\Delta_{\mathbf{y}_{j,i}} = \mathbf{y}_j - \mathbf{y}_i$ and $\Delta_{\mathbf{x}_{j,i}} = \mathbf{x}_j - \mathbf{x}_i$.[3] Thus, $\mathbf{y}$ is drawn from a multivariate normal distribution given by

$$p(\mathbf{y}|\mathbf{C}, \mathbf{x}, \mathbf{V}, \mathbf{G}) = \mathcal{N}(\boldsymbol{\mu}_{\mathbf{y}}, \boldsymbol{\Sigma}_{\mathbf{y}}),$$

with $\boldsymbol{\Sigma}_{\mathbf{y}}^{-1} = (\epsilon \mathbf{1}_n \mathbf{1}_n^\top) \otimes \mathbf{I}_{d_y} + 2\mathbf{L} \otimes \mathbf{V}^{-1}$, $\boldsymbol{\mu}_{\mathbf{y}} = \boldsymbol{\Sigma}_{\mathbf{y}}\mathbf{e}$, and $\mathbf{e} = [\mathbf{e}_1^\top, \cdots, \mathbf{e}_n^\top]^\top \in \mathbb{R}^{nd_y}$; $\mathbf{e}_i = -\sum_{j=1}^n \eta_{ji}\mathbf{V}^{-1}(\mathbf{C}_j + \mathbf{C}_i)\Delta_{\mathbf{x}_{j,i}}$. For computational simplicity, we assume $\mathbf{V}^{-1} = \gamma \mathbf{I}_{d_y}$. The graphical representation of the generative process underlying the LL-LVM is given in Fig. 2.

## 3  Variational inference

Our goal is to infer the latent variables $(\mathbf{x}, \mathbf{C})$ as well as the parameters $\boldsymbol{\theta} = \{\alpha, \gamma\}$ in LL-LVM. We infer them by maximising the lower bound $\mathcal{L}$ of the marginal likelihood of the observations

$$\log p(\mathbf{y}|\mathbf{G}, \boldsymbol{\theta}) \geq \iint q(\mathbf{C}, \mathbf{x}) \log \frac{p(\mathbf{y}, \mathbf{C}, \mathbf{x}|\mathbf{G}, \boldsymbol{\theta})}{q(\mathbf{C}, \mathbf{x})}\mathrm{d}\mathbf{x}\mathrm{d}\mathbf{C} := \mathcal{L}(q(\mathbf{C}, \mathbf{x}), \boldsymbol{\theta}).$$

(6)

Following the common treatment for computational tractability, we assume the posterior over $(\mathbf{C}, \mathbf{x})$ factorises as $q(\mathbf{C}, \mathbf{x}) = q(\mathbf{x})q(\mathbf{C})$ [13]. We maximise the lower bound w.r.t. $q(\mathbf{C}, \mathbf{x})$ and $\boldsymbol{\theta}$ by the variational expectation maximization algorithm [14], which consists of (1) the variational expectation step for computing $q(\mathbf{C}, \mathbf{x})$ by

$$q(\mathbf{x}) \propto \exp\left[\int q(\mathbf{C}) \log p(\mathbf{y}, \mathbf{C}, \mathbf{x}|\mathbf{G}, \boldsymbol{\theta})\mathrm{d}\mathbf{C}\right],$$

(7)

$$q(\mathbf{C}) \propto \exp\left[\int q(\mathbf{x}) \log p(\mathbf{y}, \mathbf{C}, \mathbf{x}|\mathbf{G}, \boldsymbol{\theta})\mathrm{d}\mathbf{x}\right],$$

(8)

then (2) the maximization step for estimating $\boldsymbol{\theta}$ by $\hat{\boldsymbol{\theta}} = \arg\max_{\boldsymbol{\theta}} \mathcal{L}(q(\mathbf{C}, \mathbf{x}), \boldsymbol{\theta})$.

**Variational-E step**  Computing $q(\mathbf{x})$ from Eq. (7) requires rewriting the likelihood in Eq. (5) as a quadratic function in $\mathbf{x}$

$$p(\mathbf{y}|\mathbf{C}, \mathbf{x}, \boldsymbol{\theta}, \mathbf{G}) = \frac{1}{\tilde{Z}_{\mathbf{x}}}\exp\left[-\frac{1}{2}(\mathbf{x}^\top \mathbf{A}\mathbf{x} - 2\mathbf{x}^\top \mathbf{b})\right],$$

where the normaliser $\tilde{Z}_{\mathbf{x}}$ has all the terms that do not depend on $\mathbf{x}$ from Eq. (5). Let $\tilde{\mathbf{L}} := (\epsilon \mathbf{1}_n \mathbf{1}_n^\top + 2\gamma \mathbf{L})^{-1}$. The matrix $\mathbf{A}$ is given by $\mathbf{A} := \mathbf{A}_E^\top \boldsymbol{\Sigma}_{\mathbf{y}} \mathbf{A}_E = [\mathbf{A}_{ij}]_{i,j=1}^n \in \mathbb{R}^{nd_x \times nd_x}$ where the $i,j$th $d_x \times d_x$ block is $\mathbf{A}_{ij} = \sum_{p=1}^n \sum_{q=1}^n \tilde{\mathbf{L}}(p,q)\mathbf{A}_E(p,i)^\top \mathbf{A}_E(q,j)$ and each $i,j$th $(d_y \times d_x)$ block of $\mathbf{A}_E \in \mathbb{R}^{nd_y \times nd_x}$ is given by $\mathbf{A}_E(i,j) = -\eta_{ij}\mathbf{V}^{-1}(\mathbf{C}_j + \mathbf{C}_i) + \delta_{ij}\left[\sum_k \eta_{ik}\mathbf{V}^{-1}(\mathbf{C}_k + \mathbf{C}_i)\right]$. The vector $\mathbf{b}$ is defined as $\mathbf{b} = [\mathbf{b}_1^\top, \cdots, \mathbf{b}_n^\top]^\top \in \mathbb{R}^{nd_x}$ with the component $d_x$-dimensional vectors given by $\mathbf{b}_i = \sum_{j=1}^n \eta_{ij}(\mathbf{C}_j^\top \mathbf{V}^{-1}(\mathbf{y}_i - \mathbf{y}_j) - \mathbf{C}_i^\top \mathbf{V}^{-1}(\mathbf{y}_j - \mathbf{y}_i))$. The likelihood combined with the prior on $\mathbf{x}$ gives us the Gaussian posterior over $\mathbf{x}$ (i.e., solving Eq. (7))

$$q(\mathbf{x}) = \mathcal{N}(\mathbf{x}|\boldsymbol{\mu}_{\mathbf{x}}, \boldsymbol{\Sigma}_{\mathbf{x}}), \quad \text{where } \boldsymbol{\Sigma}_{\mathbf{x}}^{-1} = \langle \mathbf{A} \rangle_{q(\mathbf{C})} + \boldsymbol{\Pi}^{-1}, \quad \boldsymbol{\mu}_{\mathbf{x}} = \boldsymbol{\Sigma}_{\mathbf{x}}\langle \mathbf{b} \rangle_{q(\mathbf{C})}.$$

(9)

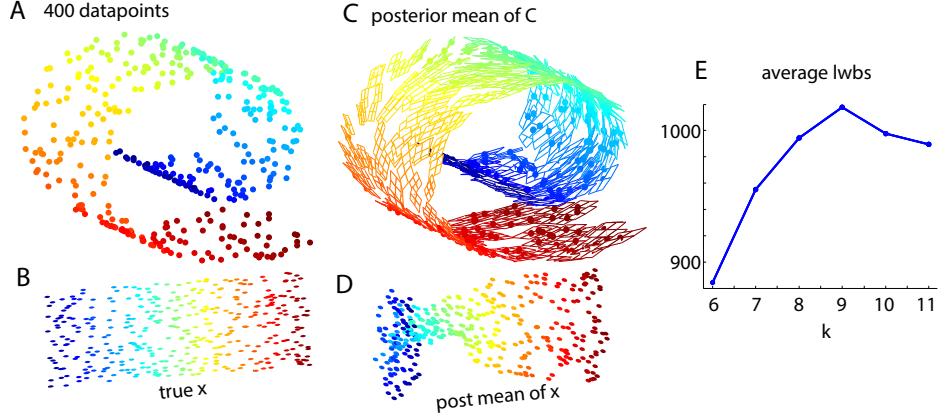

Figure 3: A simulated example. **A**: 400 data points drawn from Swiss Roll. **B**: true latent points ($\mathbf{x}$) in 2D used for generating the data. **C**: Posterior mean of $\mathbf{C}$ and **D**: posterior mean of $\mathbf{x}$ after 50 EM iterations given $k = 9$, which was chosen by maximising the lower bound across different $k$'s. **E**: Average lower bounds as a function of $k$. Each point is an average across 10 random seeds.

Similarly, computing $q(\mathbf{C})$ from Eq. (8) requires rewriting the likelihood in Eq. (5) as a quadratic function in $\mathbf{C}$

$$p(\mathbf{y}|\mathbf{C}, \mathbf{x}, \mathbf{G}, \boldsymbol{\theta}) = \frac{1}{\tilde{Z}_C} \exp[-\tfrac{1}{2}\text{Tr}(\boldsymbol{\Gamma}\mathbf{C}^\top\mathbf{C} - 2\mathbf{C}^\top\mathbf{V}^{-1}\mathbf{H})], \qquad (10)$$

where the normaliser $\tilde{Z}_C$ has all the terms that do not depend on $\mathbf{C}$ from Eq. (5), and $\boldsymbol{\Gamma} := \mathbf{Q}\tilde{\mathbf{L}}\mathbf{Q}^\top$. The matrix $\mathbf{Q} = [\mathbf{q}_1 \ \mathbf{q}_2 \ \cdots \ \mathbf{q}_n] \in \mathbb{R}^{nd_x \times n}$ where the $j$th subvector of the $i$th column is $\mathbf{q}_i(j) = \eta_{ij}\mathbf{V}^{-1}(\mathbf{x}_i - \mathbf{x}_j) + \delta_{ij}\left[\sum_k \eta_{ik}\mathbf{V}^{-1}(\mathbf{x}_i - \mathbf{x}_k)\right] \in \mathbb{R}^{d_x}$. We define $\mathbf{H} = [\mathbf{H}_1, \cdots, \mathbf{H}_n] \in \mathbb{R}^{d_y \times nd_x}$ whose $i$th block is $\mathbf{H}_i = \sum_{j=1}^{n} \eta_{ij}(\mathbf{y}_j - \mathbf{y}_i)(\mathbf{x}_j - \mathbf{x}_i)^\top$.

The likelihood combined with the prior on $\mathbf{C}$ gives us the Gaussian posterior over $\mathbf{C}$ (i.e., solving Eq. (8))

$$q(\mathbf{C}) = \mathcal{MN}(\boldsymbol{\mu}_C, \mathbf{I}, \boldsymbol{\Sigma}_C), \text{ where } \boldsymbol{\Sigma}_C^{-1} := \langle\boldsymbol{\Gamma}\rangle_{q(\mathbf{x})} + \epsilon\mathbf{J}\mathbf{J}^\top + \boldsymbol{\Omega}^{-1} \text{ and } \boldsymbol{\mu}_C = \mathbf{V}^{-1}\langle\mathbf{H}\rangle_{q(\mathbf{x})}\boldsymbol{\Sigma}_C^\top. \quad (11)$$

The expected values of $\mathbf{A}, \mathbf{b}, \boldsymbol{\Gamma}$ and $\mathbf{H}$ are given in the Appendix.

**Variational-M step**   We set the parameters by maximising $\mathcal{L}(q(\mathbf{C}, \mathbf{x}), \boldsymbol{\theta})$ w.r.t. $\boldsymbol{\theta}$ which is split into two terms based on dependence on each parameter: (1) expected log-likelihood for updating $\mathbf{V}$ by $\arg\max_{\mathbf{V}} \mathbb{E}_{q(\mathbf{x})q(\mathbf{C})}[\log p(\mathbf{y}|\mathbf{C}, \mathbf{x}, \mathbf{V}, \mathbf{G})]$; and (2) negative KL divergence between the prior and the posterior on $\mathbf{x}$ for updating $\alpha$ by $\arg\max_{\alpha} \mathbb{E}_{q(\mathbf{x})q(\mathbf{C})}[\log p(\mathbf{x}|\mathbf{G}, \alpha) - \log q(\mathbf{x})]$. The update rules for each hyperparameter are given in the Appendix.

The full EM algorithm[4] starts with an initial value of $\boldsymbol{\theta}$. In the E-step, given $q(\mathbf{C})$, compute $q(\mathbf{x})$ as in Eq. (9). Likewise, given $q(\mathbf{x})$, compute $q(\mathbf{C})$ as in Eq. (11). The parameters $\boldsymbol{\theta}$ are updated in the M-step by maximising Eq. (6). The two steps are repeated until the variational lower bound in Eq. (6) saturates. To give a sense of how the algorithm works, we visualise fitting results for a simulated example in Fig. 3. Using the graph constructed from 3D observations given different $k$, we run our EM algorithm. The posterior means of $\mathbf{x}$ and $\mathbf{C}$ given the optimal $k$ chosen by the maximum lower bound resemble the true manifolds in 2D and 3D spaces, respectively.

**Out-of-sample extension**   In the LL-LVM model one can formulate a computationally efficient out-of-sample extension technique as follows. Given $n$ data points denoted by $\mathcal{D} = \{\mathbf{y}_1, \cdots, \mathbf{y}_n\}$, the variational EM algorithm derived in the previous section converts $\mathcal{D}$ into the posterior $q(\mathbf{x}, \mathbf{C})$: $\mathcal{D} \mapsto q(\mathbf{x})q(\mathbf{C})$. Now, given a new high-dimensional data point $\mathbf{y}^*$, one can first find the neighbourhood of $\mathbf{y}^*$ without changing the current neighbourhood graph. Then, it is possible to compute the distributions over the corresponding locally linear map and latent variable $q(\mathbf{C}^*, \mathbf{x}^*)$ via simply performing the E-step given $q(\mathbf{x})q(\mathbf{C})$ (freezing all other quantities the same) as $\mathcal{D} \cup \{\mathbf{y}^*\} \mapsto q(\mathbf{x})q(\mathbf{C})q(\mathbf{x}^*)q(\mathbf{C}^*)$.

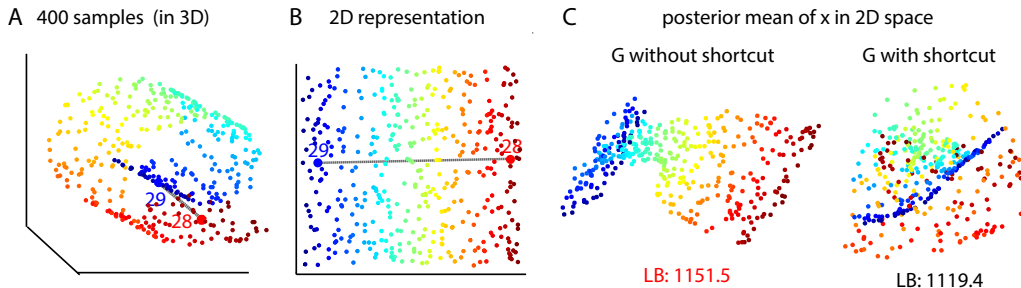

Figure 4: Resolving short-circuiting problems using variational lower bound. **A**: Visualization of 400 samples drawn from a Swiss Roll in 3D space. Points 28 (red) and 29 (blue) are close to each other (dotted grey) in 3D. **B**: Visualization of the 400 samples on the latent 2D manifold. The distance between points 28 and 29 is seen to be large. **C**: Posterior mean of $\mathbf{x}$ with/without short-circuiting the 28th and the 29th data points in the graph construction. LLLVM achieves a higher lower bound when the shortcut is absent. The red and blue parts are mixed in the resulting estimate in 2D space (right) when there is a shortcut. The lower bound is obtained after 50 EM iterations.

**Comparison to GP-LVM**   A closely related probabilistic dimensionality reduction algorithm to LL-LVM is GP-LVM [7]. GP-LVM defines the mapping from the latent space to data space using Gaussian processes. The likelihood of the observations $\mathbf{Y} = [\mathbf{y}_1, \ldots, \mathbf{y}_{d_y}] \in \mathbb{R}^{n \times d_y}$ ($\mathbf{y}_k$ is the vector formed by the $k$th element of all $n$ high dimensional vectors) given latent variables $\mathbf{X} = [\mathbf{x}_1, \ldots, \mathbf{x}_{d_x}] \in \mathbb{R}^{n \times d_x}$ is defined by $p(\mathbf{Y}|\mathbf{X}) = \prod_{k=1}^{d_y} \mathcal{N}(\mathbf{y}_k | \mathbf{0}, \mathbf{K}_{nn} + \beta^{-1} \mathbf{I}_n)$, where the $i, j$th element of the covariance matrix is of the exponentiated quadratic form: $k(\mathbf{x}_i, \mathbf{x}_j) = \sigma_f^2 \exp\left[-\frac{1}{2} \sum_{q=1}^{d_x} \alpha_q (x_{i,q} - x_{j,q})^2\right]$ with smoothness-scale parameters $\{\alpha_q\}$ [8]. In LL-LVM, once we integrate out $\mathbf{C}$ from Eq. (5), we also obtain the Gaussian likelihood given $\mathbf{x}$,

$$p(\mathbf{y}|\mathbf{x}, \mathbf{G}, \boldsymbol{\theta}) = \int p(\mathbf{y}|\mathbf{C}, \mathbf{x}, \mathbf{G}, \boldsymbol{\theta}) p(\mathbf{C}|\mathbf{G}, \boldsymbol{\theta}) \, \mathrm{d}\mathbf{C} = \tfrac{1}{Z_{Y_y}} \exp\left[-\tfrac{1}{2} \mathbf{y}^\top \mathbf{K}_{LL}^{-1} \mathbf{y}\right].$$

In contrast to GP-LVM, the precision matrix $\mathbf{K}_{LL}^{-1} = (2\mathbf{L} \otimes \mathbf{V}^{-1}) - (\mathbf{W} \otimes \mathbf{V}^{-1}) \boldsymbol{\Lambda} (\mathbf{W}^\top \otimes \mathbf{V}^{-1})$ depends on the graph Laplacian matrix through $\mathbf{W}$ and $\boldsymbol{\Lambda}$. Therefore, in LL-LVM, the *graph structure* directly determines the functional form of the conditional precision.

## 4   Experiments

### 4.1   Mitigating the short-circuit problem

Like other neighbour-based methods, LL-LVM is sensitive to misspecified neighbourhoods; the prior, likelihood, and posterior all depend on the assumed graph. Unlike other methods, LL-LVM provides a natural way to evaluate possible short-circuits using the variational lower bound of Eq. (6). Fig. 4 shows 400 samples drawn from a Swiss Roll in 3D space (Fig. 4**A**). Two points, labelled 28 and 29, happen to fall close to each other in 3D, but are actually far apart on the latent (2D) surface (Fig. 4**B**). A k-nearest-neighbour graph might link these, distorting the recovered coordinates. However, evaluating the model without this edge (the correct graph) yields a higher variational bound (Fig. 4**C**). Although it is prohibitive to evaluate every possible graph in this way, the availability of a principled criterion to test specific hypotheses is of obvious value.

In the following, we demonstrate LL-LVM on two real datasets: handwritten digits and climate data.

### 4.2   Modelling USPS handwritten digits

As a first real-data example, we test our method on a subset of 80 samples each of the digits $0, 1, 2, 3, 4$ from the USPS digit dataset, where each digit is of size $16 \times 16$ (i.e., $n = 400$, $d_y = 256$). We follow [7], and represent the low-dimensional latent variables in 2D.

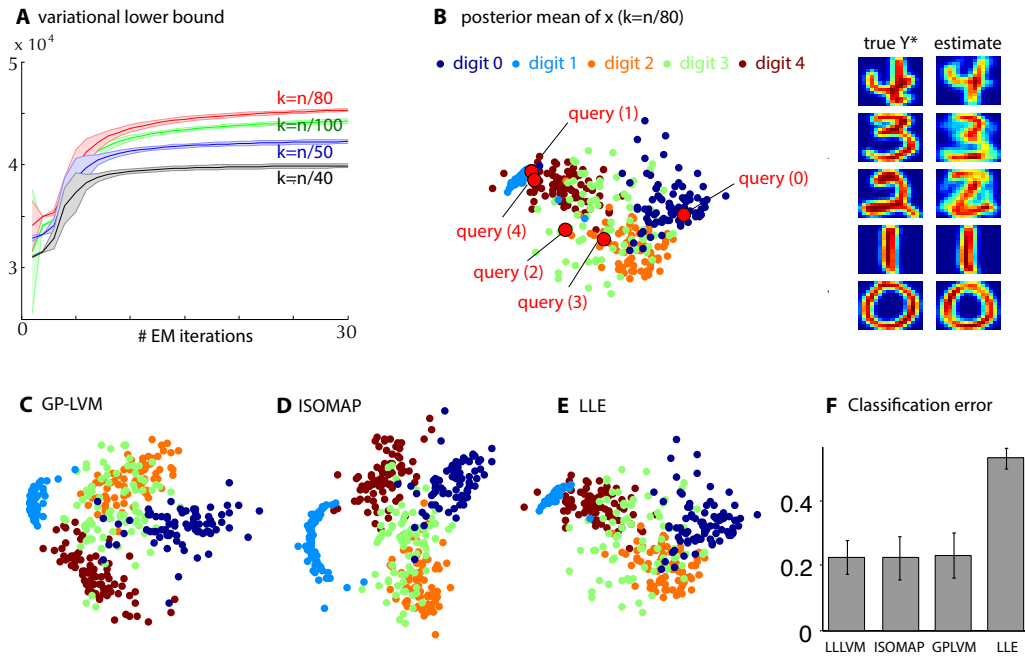

Figure 5: USPS handwritten digit dataset described in section 4.2. **A**: Mean (in solid) and variance (1 standard n deviation shading) of the variational lower bound across 10 different random starts of EM algorithm with different $k$'s. The highest lower bound is achieved when $k = n/80$. **B**: The posterior mean of $\mathbf{x}$ in 2D. Each digit is colour coded. On the right side are reconstructions of $\mathbf{y}^*$ for randomly chosen query points $\mathbf{x}^*$. Using neighbouring $\mathbf{y}$ and posterior means of $\mathbf{C}$ we can recover $\mathbf{y}^*$ successfully (see text). **C**: Fitting results by GP-LVM using the same data. **D**: ISOMAP ($k = 30$) and **E**: LLE ($k$=40). Using the extracted features (in 2D), we evaluated a 1-NN classifier for digit identity with 10-fold cross-validation (the same data divided into 10 training and test sets). The classification error is shown in **F**. LL-LVM features yield the comparably low error with GP-LVM and ISOMAP.

Fig. 5**A** shows variational lower bounds for different values of $k$, using 9 different EM initialisations. The posterior mean of $\mathbf{x}$ obtained from LL-LVM using the best $k$ is illustrated in Fig. 5**B**. Fig. 5**B** also shows reconstructions of one randomly-selected example of each digit, using its 2D coordinates $\mathbf{x}^*$ as well as the posterior mean coordinates $\hat{\mathbf{x}}_i$, tangent spaces $\hat{\mathbf{C}}_i$ and actual images $\mathbf{y}_i$ of its $k = n/80$ closest neighbours. The reconstruction is based on the assumed tangent-space structure of the generative model (Eq. (5)), that is: $\hat{\mathbf{y}}^* = \frac{1}{k} \sum_{i=1}^{k} \left[ \mathbf{y}_i + \hat{\mathbf{C}}_i(\mathbf{x}^* - \hat{\mathbf{x}}_i) \right]$. A similar process could be used to reconstruct digits at out-of-sample locations. Finally, we quantify the relevance of the recovered subspace by computing the error incurred using a simple classifier to report digit identity using the 2D features obtained by LL-LVM and various competing methods (Fig. 5**C-F**). Classification with LL-LVM coordinates performs similarly to GP-LVM and ISOMAP ($k = 30$), and outperforms LLE ($k = 40$).

## 4.3 Mapping climate data

In this experiment, we attempted to recover 2D geographical relationships between weather stations from recorded monthly precipitation patterns. Data were obtained by averaging month-by-month annual precipitation records from 2005–2014 at 400 weather stations scattered across the US (see Fig. 6) [5]. Thus, the data set comprised 400 12-dimensional vectors. The goal of the experiment is to recover the two-dimensional topology of the weather stations (as given by their latitude and longi-

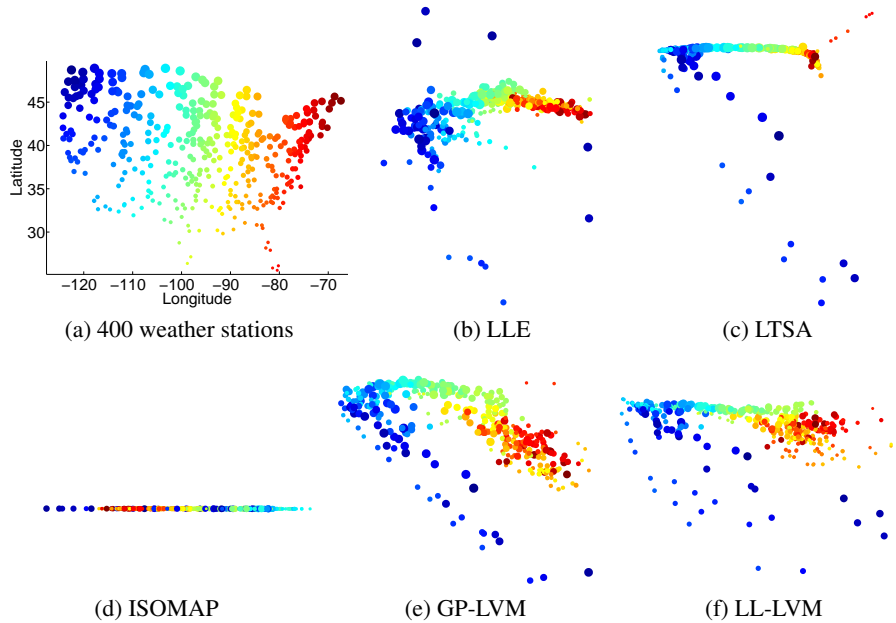

(a) 400 weather stations  (b) LLE  (c) LTSA

(d) ISOMAP  (e) GP-LVM  (f) LL-LVM

Figure 6: Climate modelling problem as described in section 4.3. Each example corresponding to a weather station is a 12-dimensional vector of monthly precipitation measurements. Using only the measurements, the projection obtained from the proposed LL-LVM recovers the topological arrangement of the stations to a large degree.

tude) using only these 12-dimensional climatic measurements. As before, we compare the projected points obtained by LL-LVM with several widely used dimensionality reduction techniques. For the graph-based methods LL-LVM, LTSA, ISOMAP, and LLE, we used 12-NN with Euclidean distance to construct the neighbourhood graph.

The results are presented in Fig. 6. LL-LVM identified a more geographically-accurate arrangement for the weather stations than the other algorithms. The fully probabilistic nature of LL-LVM and GPLVM allowed these algorithms to handle the noise present in the measurements in a principled way. This contrasts with ISOMAP which can be topologically unstable [16] i.e. vulnerable to short-circuit errors if the neighbourhood is too large. Perhaps coincidentally, LL-LVM also seems to respect local geography more fully in places than does GP-LVM.

## 5   Conclusion

We have demonstrated a new probabilistic approach to non-linear manifold discovery that embodies the central notion that local geometries are mapped linearly between manifold coordinates and high-dimensional observations. The approach offers a natural variational algorithm for learning, quantifies local uncertainty in the manifold, and permits evaluation of hypothetical neighbourhood relationships.

In the present study, we have described the LL-LVM model conditioned on a neighbourhood graph. In principle, it is also possible to extend LL-LVM so as to construct a distance matrix as in [17], by maximising the data likelihood. We leave this as a direction for future work.

### Acknowledgments

The authors were funded by the Gatsby Charitable Foundation.

## Footnotes

[1]This is also true of one previous MFA-based method [12] which finds model parameters and global coordinates by variational methods similar to our own.

[2] $\epsilon$ sets the scale of the average linear map, ensuring the prior precision matrix is invertible.

[3]The $\epsilon$ term centers the data and ensures the distribution can be normalised. It applies in a subspace orthogonal to that modelled by $\mathbf{x}$ and $\mathbf{C}$ and so its value does not affect the resulting manifold model.

[4]An implementation is available from `http://www.gatsby.ucl.ac.uk/resources/lllvm`.

[5]The dataset is made available by the National Climatic Data Center at `http://www.ncdc.noaa.gov/oa/climate/research/ushcn/`. We use version 2.5 monthly data [15].

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
