[Supplementary Material]

# LL-LVM supplementary material

**Notation**   The vectorized version of a matrix is vec($\mathbf{M}$). We denote an identity matrix of size $m$ with $\mathbf{I}_m$. Other notations are the same as used in the main text.

## A   Matrix normal distribution

The matrix normal distribution generalises the standard multivariate normal distribution to matrix-valued variables. A matrix $\mathbf{A} \in \mathbb{R}^{n \times p}$ is said to follow a matrix normal distribution $\mathcal{MN}_{n,p}(\mathbf{M}, \mathbf{U}, \mathbf{V})$ with parameters $\mathbf{U}$ and $\mathbf{V}$ if its density is given by

$$p(\mathbf{A} \mid \mathbf{M}, \mathbf{U}, \mathbf{V}) = \frac{\exp\left(-\frac{1}{2} \operatorname{Tr}\left[\mathbf{V}^{-1}(\mathbf{A} - \mathbf{M})^T \mathbf{U}^{-1}(\mathbf{A} - \mathbf{M})\right]\right)}{(2\pi)^{np/2}|\mathbf{V}|^{n/2}|\mathbf{U}|^{p/2}}. \tag{1}$$

If $\mathbf{A} \sim \mathcal{MN}(\mathbf{M}, \mathbf{U}, \mathbf{V})$, then vec($\mathbf{A}$) $\sim \mathcal{N}(\text{vec}(\mathbf{M}), \mathbf{V} \otimes \mathbf{U})$, a relationship we will use to simplify many expressions.

## B   Matrix normal expressions of priors and likelihood

Recall that $\mathbf{G}_{ij} = \eta_{ij}$.

**Prior on low dimensional latent variables**

$$\log p(\mathbf{x}|\mathbf{G}, \alpha) = -\frac{\alpha}{2} \sum_{i=1}^{n} ||\mathbf{x}_i||^2 - \frac{1}{2} \sum_{i=1}^{n} \sum_{j=1}^{n} \eta_{ij} ||\mathbf{x}_i - \mathbf{x}_j||^2 - \log Z_{\mathbf{x}} \tag{2}$$

$$= -\frac{1}{2} \log |2\pi\mathbf{\Pi}| - \frac{1}{2} \mathbf{x}^\top \mathbf{\Pi}^{-1} \mathbf{x}, \tag{3}$$

where

$$\mathbf{\Pi}^{-1} := \alpha \mathbf{I}_{nd_x} + \mathbf{\Omega}^{-1},$$
$$\mathbf{\Omega}^{-1} := 2\mathbf{L} \otimes \mathbf{I}_{d_x},$$
$$\mathbf{L} := \operatorname{diag}(\mathbf{G1}) - \mathbf{G}.$$

$\mathbf{L}$ is known as a graph Laplacian. It follows that $p(\mathbf{x}|\mathbf{G}, \alpha) = \mathcal{N}(\mathbf{0}, \mathbf{\Pi})$. The prior covariance $\mathbf{\Pi}$ can be rewritten as

$$\mathbf{\Pi}^{-1} = \alpha \mathbf{I}_n \otimes \mathbf{I}_{d_x} + 2\mathbf{L} \otimes \mathbf{I}_{d_x} \tag{4}$$
$$= (\alpha \mathbf{I}_n + 2\mathbf{L}) \otimes \mathbf{I}_{d_x}, \tag{5}$$
$$\mathbf{\Pi} = (\alpha \mathbf{I}_n + 2\mathbf{L})^{-1} \otimes \mathbf{I}_{d_x}. \tag{6}$$

By the relationship of a matrix normal and multivariate normal distributions described in section A, the equivalent prior for the matrix $\mathbf{X} = [\mathbf{x}_1 \mathbf{x}_2 \cdots \mathbf{x}_n] \in \mathbb{R}^{d_x \times n}$, constructed by reshaping $\mathbf{x}$, is given by

$$p(\mathbf{X}|\mathbf{G}, \alpha) = \mathcal{MN}(\mathbf{X}|\mathbf{0}, \mathbf{I}_{d_x}, (\alpha \mathbf{I}_n + 2\mathbf{L})^{-1}). \tag{7}$$

**Prior on locally linear maps**

Recall that $\mathbf{C} = [\mathbf{C}_1, \ldots, \mathbf{C}_n] \in \mathbb{R}^{d_y \times nd_x}$ where each $\mathbf{C}_i \in \mathbb{R}^{d_y \times d_x}$. We formulate the log prior on $\mathbf{C}$ as

$$\log p(\mathbf{C}|\mathbf{G}) = -\frac{\epsilon}{2} || \sum_{i=1}^{n} \mathbf{C}_i ||_F^2 - \frac{1}{2} \sum_{i=1}^{n} \sum_{j=1}^{n} \eta_{ij} ||\mathbf{C}_i - \mathbf{C}_j||_F^2 - \log Z_{\mathbf{c}},$$

$$= -\frac{\epsilon}{2} \operatorname{Tr}\left(\mathbf{C}\mathbf{J}\mathbf{J}^\top \mathbf{C}^\top\right) - \frac{1}{2} \operatorname{Tr}\left(\mathbf{\Omega}^{-1}\mathbf{C}^\top \mathbf{C}\right) - \log Z_{\mathbf{c}}, \text{ where } \mathbf{J} := \mathbf{1}_n \otimes \mathbf{I}_{d_x},$$

$$= -\frac{1}{2} \operatorname{Tr}\left[(\epsilon \mathbf{J}\mathbf{J}^\top + \mathbf{\Omega}^{-1})\mathbf{C}^\top \mathbf{C}\right] - \log Z_{\mathbf{c}}. \tag{8}$$

In the first line, the first term imposes a constraint that the mean of $\mathbf{C}_i$ should not be too large. The second term encourages the the locally linear maps of neighbouring points $i$ and $j$ to be similar in the sense of the Frobenius norm. Notice that the last line is in the form of a the log of a matrix normal density with mean 0 where $Z_\mathbf{c}$ is given by

$$\log Z_\mathbf{c} = \frac{nd_xd_y}{2}\log|2\pi| - \frac{d_y}{2}\log|\epsilon\mathbf{J}\mathbf{J}^\top + \mathbf{\Omega}^{-1}| \tag{9}$$

The expression is equivalent to

$$p(\mathbf{C}|\mathbf{G}) = \mathcal{MN}(\mathbf{C}|\mathbf{0}, \mathbf{I}_{d_y}, (\epsilon\mathbf{J}\mathbf{J}^\top + \mathbf{\Omega}^{-1})^{-1}). \tag{10}$$

In our implementation, we fix $\epsilon$ to a small value, since the magnitude of $\mathbf{C}_i$ and $\mathbf{x}_i$ can be controlled by the hyper-parameter $\alpha$, which is optimized in the M-step.

**Likelihood**

We penalise linear approximation error of the tangent spaces. Assume that the noise precision matrix is a scaled identify matrix ei.g., $\mathbf{V}^{-1} = \gamma\mathbf{I}_{d_y}$.

$$\log p(\mathbf{y}|\mathbf{x}, \mathbf{C}, \mathbf{V}, \mathbf{G}) = -\frac{\epsilon}{2}||\sum_{i=1}^n \mathbf{y}_i||^2 - \log Z_\mathbf{y} \tag{11}$$

$$-\frac{1}{2}\sum_{i=1}^n\sum_{j=1}^n \eta_{ij}((\mathbf{y}_j - \mathbf{y}_i) - \mathbf{C}_i(\mathbf{x}_j - \mathbf{x}_i))^\top\mathbf{V}^{-1}((\mathbf{y}_j - \mathbf{y}_i) - \mathbf{C}_i(\mathbf{x}_j - \mathbf{x}_i)),$$

$$= -\frac{1}{2}(\mathbf{y}^\top\Sigma_\mathbf{y}^{-1}\mathbf{y} - 2\mathbf{y}^\top\mathbf{e} + f) - \log Z_\mathbf{y}, \tag{12}$$

where

$$\mathbf{y} = [\mathbf{y}_1^\top, \cdots, \mathbf{y}_n^\top]^\top \in \mathbb{R}^{nd_y} \tag{13}$$

$$\Sigma_\mathbf{y}^{-1} = (\epsilon\mathbf{1}_n\mathbf{1}_n^\top)\otimes\mathbf{I}_{d_y} + 2\mathbf{L}\otimes\mathbf{V}^{-1}, \tag{14}$$

$$\mathbf{e} = [\mathbf{e}_1^\top, \cdots, \mathbf{e}_n^\top]^\top \in R^{nd_y}, \tag{15}$$

$$\mathbf{e}_i = -\sum_{j=1}^n \eta_{ji}\mathbf{V}^{-1}(\mathbf{C}_j + \mathbf{C}_i)(\mathbf{x}_j - \mathbf{x}_i), \tag{16}$$

$$f = \sum_{i=1}^n\sum_{j=1}^n \eta_{ij}(\mathbf{x}_j - \mathbf{x}_i)^\top\mathbf{C}_i^\top\mathbf{V}^{-1}\mathbf{C}_i(\mathbf{x}_j - \mathbf{x}_i). \tag{17}$$

By completing the quadratic form in $\mathbf{y}$, we want to write down the likelihood as a multivariate Gaussian [1] :

$$p(\mathbf{y}|\mathbf{x}, \mathbf{C}, \mathbf{V}, \mathbf{G}) = \mathcal{N}(\boldsymbol{\mu}_\mathbf{y}, \Sigma_\mathbf{y}), \tag{18}$$

$$\boldsymbol{\mu}_\mathbf{y} = \Sigma_\mathbf{y}\mathbf{e}. \tag{19}$$

$$p(\mathbf{Y}|\mathbf{x}, \mathbf{C}, \gamma, \mathbf{G}) = \mathcal{MN}(\mathbf{Y}|\mathbf{M}_\mathbf{y}, \mathbf{I}_{d_y}, (\epsilon\mathbf{1}_n\mathbf{1}_n^\top + 2\gamma\mathbf{L})^{-1}),$$

$$\mathbf{M}_\mathbf{y} = \mathbf{E}(\epsilon\mathbf{1}_n\mathbf{1}_n^\top + 2\gamma\mathbf{L})^{-1},$$

where $\mathbf{E} = [\mathbf{e}_1, \cdots, \mathbf{e}_n] \in R^{d_y \times n}$. The covariance in Eq. (13) decomposes

$$\Sigma_\mathbf{y}^{-1} = (\epsilon\mathbf{1}_n\mathbf{1}_n^\top)\otimes\mathbf{I}_{d_y} + 2\mathbf{L}\otimes\mathbf{V}^{-1},$$

$$= (\epsilon\mathbf{1}_n\mathbf{1}_n^\top + 2\gamma\mathbf{L})\otimes\mathbf{I}_{d_y},$$

$$\Sigma_\mathbf{y} = (\epsilon\mathbf{1}_n\mathbf{1}_n^\top + 2\gamma\mathbf{L})^{-1}\otimes\mathbf{I}_{d_y}.$$

By equating Eq. (11) with Eq. (18), we get the normalisation term $Z_{\mathbf{y}}$

$$-\frac{1}{2}(\mathbf{y}^\top \Sigma_{\mathbf{y}}^{-1}\mathbf{y} - 2\mathbf{y}^\top \mathbf{e} + f) - \log Z_{\mathbf{y}} = -\frac{1}{2}(\mathbf{y} - \boldsymbol{\mu_y})^\top \Sigma_{\mathbf{y}}^{-1}(\mathbf{y} - \boldsymbol{\mu_y}) - \frac{1}{2}\log|2\pi\Sigma_{\mathbf{y}}|, \tag{20}$$

$$\log Z_{\mathbf{y}} = \frac{1}{2}(\boldsymbol{\mu_y}^\top \Sigma_{\mathbf{y}}^{-1}\boldsymbol{\mu_y} - f) + \frac{1}{2}\log|2\pi\Sigma_{\mathbf{y}}|, \tag{21}$$

$$Z_{\mathbf{y}} = \exp(\tfrac{1}{2}(\boldsymbol{\mu_y}^\top \Sigma_{\mathbf{y}}^{-1}\boldsymbol{\mu_y} - f))|2\pi\Sigma_{\mathbf{y}}|^{\frac{1}{2}}, \tag{22}$$

$$= \exp(\tfrac{1}{2}(\mathbf{e}^\top \Sigma_{\mathbf{y}}\mathbf{e} - f))|2\pi\Sigma_{\mathbf{y}}|^{\frac{1}{2}}. \tag{23}$$

Therefore, the normalised log-likelihood can be written as

$$\log p(\mathbf{y}|\mathbf{x}, \mathbf{C}, \mathbf{V}, \mathbf{G}) = -\frac{1}{2}(\mathbf{y}^\top \Sigma_{\mathbf{y}}^{-1}\mathbf{y} - 2\mathbf{y}^\top \mathbf{e} + \mathbf{e}^\top \Sigma_{\mathbf{y}}\mathbf{e}) - \frac{1}{2}\log|2\pi\Sigma_{\mathbf{y}}|. \tag{24}$$

**Convenient form for EM**

For the EM derivation in the next section, it is convenient to write the exponent term in terms of linear and quadratic functions in $\mathbf{x}$ and $\mathbf{C}$, respectively. The linear terms appear in $\mathbf{y}^\top \mathbf{e}$, which we write as a linear function in $\mathbf{x}$ or $\mathbf{C}$

$$\mathbf{y}^\top \mathbf{e} = \mathbf{x}^\top \mathbf{b}, \tag{25}$$

$$= \mathrm{Tr}(\mathbf{C}^\top \mathbf{V}^{-1}\mathbf{H}), \tag{26}$$

where

$$\mathbf{H} = [\mathbf{H}_1, \cdots, \mathbf{H}_n] \in R^{d_y \times nd_x}, \quad \text{where } \mathbf{H}_i = \sum_{j=1}^{n} \eta_{ij}(\mathbf{y}_j - \mathbf{y}_i)(\mathbf{x}_j - \mathbf{x}_i)^\top, \tag{27}$$

$$\mathbf{b} = [\mathbf{b}_1^\top, \cdots, \mathbf{b}_n^\top]^\top \in R^{nd_x}, \quad \text{where } \mathbf{b}_i = \sum_{j=1}^{n} \eta_{ij}(\mathbf{C}_j^\top \mathbf{V}^{-1}(\mathbf{y}_i - \mathbf{y}_j) - \mathbf{C}_i^\top \mathbf{V}^{-1}(\mathbf{y}_j - \mathbf{y}_i)). \tag{28}$$

The quadratic terms appear in $\mathbf{e}^\top \Sigma_{\mathbf{y}}\mathbf{e}$, which we write as a quadratic function of $\mathbf{x}$ or a quadratic function of $\mathbf{C}$

$$\mathbf{e}^\top \Sigma_{\mathbf{y}}\mathbf{e} = \mathbf{x}^\top \mathbf{A}_E^\top \Sigma_{\mathbf{y}} \mathbf{A}_E \mathbf{x}, \tag{29}$$

$$= \mathrm{Tr}[\mathbf{Q}\tilde{\mathbf{L}}\mathbf{Q}^\top \mathbf{C}^\top \mathbf{C}], \tag{30}$$

where the $i, j$th $(d_y \times d_x)$ chunk of $\mathbf{A}_E \in \mathbb{R}^{nd_y \times nd_x}$ is given by

$$\mathbf{A}_E(i, j) = -\eta_{ij}\mathbf{V}^{-1}(\mathbf{C}_j + \mathbf{C}_i) + \delta_{ij}\left[\sum_k \eta_{ik}\mathbf{V}^{-1}(\mathbf{C}_k + \mathbf{C}_i)\right]. \tag{31}$$

The matrix $\tilde{\mathbf{L}} = (\epsilon \mathbf{1}_n \mathbf{1}_n^\top + 2\gamma \mathbf{L})^{-1}$ and $\mathbf{Q} = [\mathbf{q}_1\ \mathbf{q}_2\ \cdots\ \mathbf{q}_n] \in \mathbb{R}^{nd_x \times n}$ and the $i$th column of this matrix is denoted by $\mathbf{q}_i \in \mathbb{R}^{nd_x}$. The $j$th chunk (of length $d_x$) of the $i$th column is given by

$$\mathbf{q}_i(j) = \eta_{ij}\mathbf{V}^{-1}(\mathbf{x}_i - \mathbf{x}_j) + \delta_{ij}\left[\sum_k \eta_{ik}\mathbf{V}^{-1}(\mathbf{x}_i - \mathbf{x}_k)\right]. \tag{32}$$

# C   Variational inference

In LL-LVM, the goal is to infer the latent variables $(\mathbf{x}, \mathbf{C})$ as well as to learn the hyper-parameters $\boldsymbol{\theta} = \{\alpha, \gamma\}$. We infer them by maximising the lower bound of the marginal likelihood of the observations $\mathbf{y}$.

$$\begin{aligned}
\log p(\mathbf{y}|\boldsymbol{\theta}, \mathbf{G}) &= \log \int \int p(\mathbf{y}, \mathbf{C}, \mathbf{x}|\mathbf{G}, \boldsymbol{\theta})\, \mathrm{d}\mathbf{x}\, \mathrm{d}\mathbf{C}, \\
&\geq \int \int \int q(\mathbf{C}, \mathbf{x})\, \log \frac{p(\mathbf{y}, \mathbf{C}, \mathbf{x}|\mathbf{G}, \boldsymbol{\theta})}{q(\mathbf{C}, \mathbf{x})} \mathrm{d}\mathbf{x}\mathrm{d}\mathbf{C}, \\
&= \mathcal{F}(q(\mathbf{C}, \mathbf{x}), \boldsymbol{\theta}).
\end{aligned}$$

For computational tractability, we assume that the posterior over $(\mathbf{C}, \mathbf{x})$ factorizes as

$$q(\mathbf{C}, \mathbf{x}) \quad = \quad q(\mathbf{x})q(\mathbf{C}). \tag{33}$$

where $q(\mathbf{x})$ and $q(\mathbf{C})$ are multivariate normal distributions.

We maximize the lower bound w.r.t. $q(\mathbf{C}, \mathbf{x})$ and $\boldsymbol{\theta}$ by the variational expectation maximization algorithm, which consists of (1) the variational expectation step for determining $q(\mathbf{C}, \mathbf{x})$ by

$$q(\mathbf{x}) \propto \exp\left[\int q(\mathbf{C}) \log p(\mathbf{y}, \mathbf{C}, \mathbf{x} | \mathbf{G}, \boldsymbol{\theta}) \mathrm{d}\mathbf{C}\right], \tag{34}$$

$$q(\mathbf{C}) \propto \exp\left[\int q(\mathbf{x}) \log p(\mathbf{y}, \mathbf{C}, \mathbf{x} | \mathbf{G}, \boldsymbol{\theta}) \mathrm{d}\mathbf{x}\right], \tag{35}$$

followed by (2) the maximization step for estimating $\boldsymbol{\theta}$, $\hat{\boldsymbol{\theta}} = \arg\max_{\boldsymbol{\theta}} \mathcal{F}(q(\mathbf{C}, \mathbf{x}), \boldsymbol{\theta})$.

## C.1 VE step

### C.1.1 Computing $q(\mathbf{x})$

In variational E-step, we compute $q(\mathbf{x})$ by integrating out $\mathbf{C}$ from the total log joint distribution:

$$\log q(\mathbf{x}) \quad = \quad \mathbb{E}_{q(\mathbf{C})}\left[\log p(\mathbf{y}, \mathbf{C}, \mathbf{x} | \mathbf{G}, \boldsymbol{\theta})\right] + const, \tag{36}$$

$$= \quad \mathbb{E}_{q(\mathbf{C})}\left[\log p(\mathbf{y} | \mathbf{C}, \mathbf{x}, \mathbf{G}, \boldsymbol{\theta}) + \log p(\mathbf{x} | \mathbf{G}, \boldsymbol{\theta}) + \log p(\mathbf{C} | \mathbf{G}, \boldsymbol{\theta})\right] + const. \tag{37}$$

To determine $q(\mathbf{x})$, we firstly re-write $p(\mathbf{y} | \mathbf{C}, \mathbf{x}, \mathbf{G}, \boldsymbol{\theta})$ as a quadratic function in $\mathbf{x}$ :

$$\log p(\mathbf{y} | \mathbf{C}, \mathbf{x}, \mathbf{G}, \boldsymbol{\theta}) = -\frac{1}{2}(\mathbf{x}^\top \mathbf{A}_E{}^\top \boldsymbol{\Sigma}_{\mathbf{y}} \mathbf{A}_E \mathbf{x} - 2\mathbf{x}^\top \mathbf{b}) + const, \tag{38}$$

where

$$\mathbf{A} := \mathbf{A}_E{}^\top \boldsymbol{\Sigma}_{\mathbf{y}} \mathbf{A}_E, \tag{39}$$

$$\mathbf{A} = \begin{bmatrix} \mathbf{A}_{11} & \mathbf{A}_{12} & \cdots & \mathbf{A}_{1n} \\ \vdots & & \ddots & \vdots \\ \mathbf{A}_{n1} & \cdots & \cdots & \mathbf{A}_{nn} \end{bmatrix} \in \mathbb{R}^{nd_x \times nd_x}, \tag{40}$$

$$\mathbf{A}_{ij} = \sum_{p=1}^{n} \sum_{q=1}^{n} \tilde{\mathbf{L}}(p, q) \mathbf{A}_E(p, i)^\top \mathbf{A}_E(q, j) \tag{41}$$

where $\tilde{\mathbf{L}} := (\epsilon \mathbf{1}_n \mathbf{1}_n^\top + 2\gamma \mathbf{L})^{-1}$. With the likelihood expressed as a quadratic function of $\mathbf{x}$, the log posterior over $\mathbf{x}$ is given by

$$\log q(\mathbf{x}) = -\frac{1}{2} \mathbb{E}_{q(\mathbf{C})}\left[\mathbf{x}^\top \mathbf{A} \mathbf{x} - 2\mathbf{x}^\top \mathbf{b} + \mathbf{x}^\top \boldsymbol{\Pi}^{-1} \mathbf{x}\right] + const, \tag{42}$$

$$= -\frac{1}{2}\left[\mathbf{x}^\top (\langle \mathbf{A} \rangle_{q(\mathbf{C})} + \boldsymbol{\Pi}^{-1})\mathbf{x} - 2\mathbf{x}^\top \langle \mathbf{b} \rangle_{q(\mathbf{C})}\right] + const, \tag{43}$$

The posterior over $\mathbf{x}$ is given by

$$q(\mathbf{x}) = \mathcal{N}(\mathbf{x} | \boldsymbol{\mu}_{\mathbf{x}}, \boldsymbol{\Sigma}_{\mathbf{x}}), \tag{44}$$

where

$$\boldsymbol{\Sigma}_{\mathbf{x}}^{-1} = \langle \mathbf{A} \rangle_{q(\mathbf{C})} + \boldsymbol{\Pi}^{-1}, \tag{45}$$

$$\boldsymbol{\mu}_{\mathbf{x}} = \boldsymbol{\Sigma}_{\mathbf{x}} \langle \mathbf{b} \rangle_{q(\mathbf{C})}. \tag{46}$$

Notice that the parameters of $q(\mathbf{x})$ depend on the sufficient statistics $\langle \mathbf{A} \rangle_{q(\mathbf{C})}$ and $\langle \mathbf{b} \rangle_{q(\mathbf{C})}$ whose explicit forms are given in section C.1.2.

### C.1.2 Sufficient statistics A and b for $q(\mathbf{x})$

Given the posterior over $\mathbf{c}$, the sufficient statistics $\langle \mathbf{A} \rangle_{q(\mathbf{C})}$ and $\langle \mathbf{b} \rangle_{q(\mathbf{C})}$ necessary to characterise $q(\mathbf{x})$ are computed as following:

$$\langle \mathbf{A}_{ij} \rangle_{q(\mathbf{c})} = \sum_{p=1}^{n} \sum_{q=1}^{n} \tilde{\mathbf{L}}(p,q) \langle \mathbf{A}_E(p,i)^\top \mathbf{A}_E(q,j) \rangle_{q(\mathbf{c})}, \tag{47}$$

$$= \gamma^2 \sum_{p=1}^{n} \sum_{q=1}^{n} \tilde{\mathbf{L}}(p,q) \langle (-\eta_{pi}(\mathbf{C}_p + \mathbf{C}_i) + \delta_{pi} \sum_k \eta_{pk}(\mathbf{C}_k + \mathbf{C}_p))^\top (-\eta_{qj}(\mathbf{C}_q + \mathbf{C}_j) + \delta_{qj} \sum_{k'} \eta_{qk'}(\mathbf{C}_{k'} + \mathbf{C}_q)) \rangle_{q(\mathbf{c})}$$

$$= \gamma^2 \sum_{p=1}^{n} \sum_{q=1}^{n} \tilde{\mathbf{L}}(p,q) (\ \eta_{pi}\eta_{qj} \langle \mathbf{C}_p^\top \mathbf{C}_q + \mathbf{C}_p^\top \mathbf{C}_j + \mathbf{C}_i^\top \mathbf{C}_q + \mathbf{C}_i^\top \mathbf{C}_j \rangle_{q(\mathbf{c})}$$

$$- \eta_{pi}\delta_{qj} \sum_{k'} \eta_{qk'} \langle \mathbf{C}_p^\top \mathbf{C}_{k'} + \mathbf{C}_p^\top \mathbf{C}_q + \mathbf{C}_i^\top \mathbf{C}_{k'} + \mathbf{C}_i^\top \mathbf{C}_q \rangle_{q(\mathbf{c})}$$

$$- \eta_{qj}\delta_{pi} \sum_{k} \eta_{pk} \langle \mathbf{C}_k^\top \mathbf{C}_q + \mathbf{C}_k^\top \mathbf{C}_j + \mathbf{C}_p^\top \mathbf{C}_q + \mathbf{C}_p^\top \mathbf{C}_j \rangle_{q(\mathbf{c})}$$

$$+ \delta_{pi}\delta_{qj} \sum_{k} \sum_{k'} \eta_{pk}\eta_{qk'} \langle \mathbf{C}_k^\top \mathbf{C}_{k'} + \mathbf{C}_k^\top \mathbf{C}_q + \mathbf{C}_p^\top \mathbf{C}_{k'} + \mathbf{C}_p^\top \mathbf{C}_q \rangle_{q(\mathbf{c})} \ )$$

Thanks to the delta function, the last three terms above are non-zero only when $p = i$ and $q = j$. Therefore, we can replace $p$ with $i$, and $q$ with $j$, which simplifies the above as

$$\gamma^2 \sum_{p=1}^{n} \sum_{q=1}^{n} \tilde{\mathbf{L}}(p,q)\ \eta_{pi}\eta_{qj} \langle \mathbf{C}_p^\top \mathbf{C}_q + \mathbf{C}_p^\top \mathbf{C}_j + \mathbf{C}_i^\top \mathbf{C}_q + \mathbf{C}_i^\top \mathbf{C}_j \rangle_{q(\mathbf{c})}$$

$$- \gamma^2 \sum_{p=1}^{n} \sum_{k'} \tilde{\mathbf{L}}(p,j) \eta_{pi}\eta_{jk'} \langle \mathbf{C}_p^\top \mathbf{C}_{k'} + \mathbf{C}_p^\top \mathbf{C}_j + \mathbf{C}_i^\top \mathbf{C}_{k'} + \mathbf{C}_i^\top \mathbf{C}_j \rangle_{q(\mathbf{c})}$$

$$- \gamma^2 \sum_{q=1}^{n} \sum_{k} \tilde{\mathbf{L}}(i,q) \eta_{qj}\eta_{ik} \langle \mathbf{C}_k^\top \mathbf{C}_q + \mathbf{C}_k^\top \mathbf{C}_j + \mathbf{C}_i^\top \mathbf{C}_q + \mathbf{C}_i^\top \mathbf{C}_j \rangle_{q(\mathbf{c})}$$

$$+ \gamma^2 \tilde{\mathbf{L}}(i,j) \sum_{k} \sum_{k'} \eta_{ik}\eta_{jk'} \langle \mathbf{C}_k^\top \mathbf{C}_{k'} + \mathbf{C}_k^\top \mathbf{C}_j + \mathbf{C}_i^\top \mathbf{C}_{k'} + \mathbf{C}_i^\top \mathbf{C}_j \rangle_{q(\mathbf{c})}$$

We can make the equation above even simpler by replacing $k'$ with $q$ (second line), $k$ with $p$ (third line), and both $k$ and $k'$ with $p$ and $q$ (fourth line), which gives us

$$\langle \mathbf{A}_{ij} \rangle_{q(\mathbf{c})} = \gamma^2 \sum_{p=1}^{n} \sum_{q=1}^{n} [\tilde{\mathbf{L}}(p,q) - \tilde{\mathbf{L}}(p,j) - \tilde{\mathbf{L}}(i,q) + \tilde{\mathbf{L}}(i,j)]\ \eta_{pi}\eta_{qj} \langle \mathbf{C}_p^\top \mathbf{C}_q + \mathbf{C}_p^\top \mathbf{C}_j + \mathbf{C}_i^\top \mathbf{C}_q + \mathbf{C}_i^\top \mathbf{C}_j \rangle_{q(\mathbf{c})}. \tag{48}$$

For $\mathbf{b}_i$, we have

$$\langle \mathbf{b}_i \rangle_{q(\mathbf{c})} = \gamma \sum_{j=1}^{n} \eta_{ij} (\langle \mathbf{C}_j \rangle_{q(\mathbf{c})}^\top (\mathbf{y}_i - \mathbf{y}_j) - \langle \mathbf{C}_i \rangle_{q(\mathbf{c})}^\top (\mathbf{y}_j - \mathbf{y}_i)), \tag{49}$$

where $\tag{50}$

$$\langle \mathbf{C}_i \rangle_{q(\mathbf{c})} = \text{i-th chunk of } \boldsymbol{\mu}_\mathbf{C}, \text{ where each chunk is } (d_y \times d_x) \tag{51}$$

$$\langle \mathbf{C}_i^\top \mathbf{C}_j \rangle_{q(\mathbf{c})} = \text{(i,j)-th}(d_x \times d_x) \text{ chunk of } dy\boldsymbol{\Sigma}_\mathbf{C} + \langle \mathbf{C}_i \rangle_{q(\mathbf{c})}^\top \langle \mathbf{C}_j \rangle_{q(\mathbf{c})}, \tag{52}$$

### C.1.3 Computing $q(\mathbf{C})$

Next, we compute $q(\mathbf{C})$ by integrating out $\mathbf{x}$ from the total log joint distribution:

$$\log q(\mathbf{C}) = \mathbb{E}_{q(\mathbf{x})} [\log p(\mathbf{y}, \mathbf{C}, \mathbf{X}|\mathbf{G}, \boldsymbol{\theta})] + const, \tag{53}$$

$$= \mathbb{E}_{q(\mathbf{x})} [\log p(\mathbf{y}|\mathbf{C}, \mathbf{x}, \mathbf{G}, \boldsymbol{\theta}) + \log p(\mathbf{x}|\mathbf{G}, \boldsymbol{\theta})] + \log p(\mathbf{C}|\mathbf{G}, \boldsymbol{\theta}) + const. \tag{54}$$

We re-write $p(\mathbf{y}|\mathbf{C}, \mathbf{x}, \mathbf{G}, \boldsymbol{\theta})$ as a quadratic function in $\mathbf{C}$:

$$\log p(\mathbf{y}|\mathbf{C}, \mathbf{x}, \mathbf{G}, \boldsymbol{\theta}) = -\frac{1}{2}\mathrm{Tr}(\mathbf{Q}\tilde{\mathbf{L}}\mathbf{Q}^\top\mathbf{C}^\top\mathbf{C} - 2\mathbf{C}^\top\mathbf{V}^{-1}\mathbf{H}) + const, \tag{55}$$

where

$$\boldsymbol{\Gamma} := \mathbf{Q}\tilde{\mathbf{L}}\mathbf{Q}^\top,$$

$$\boldsymbol{\Gamma} = \begin{bmatrix} \boldsymbol{\Gamma}_{11} & \boldsymbol{\Gamma}_{12} & \cdots & \boldsymbol{\Gamma}_{1n} \\ \vdots & & \ddots & \vdots \\ \boldsymbol{\Gamma}_{n1} & \cdots & \cdots & \boldsymbol{\Gamma}_{nn} \end{bmatrix}$$

$$\boldsymbol{\Gamma}_{ij} = \sum_{k=1}^{n}\sum_{k'=1}^{n}\tilde{\mathbf{L}}(k, k')\mathbf{q}_k(i)\mathbf{q}_{k'}(j)^\top. \tag{56}$$

The log posterior over $\mathbf{C}$ is given by

$$\log q(\mathbf{C}) = -\frac{1}{2}\mathrm{Tr}\left[\langle\boldsymbol{\Gamma}\rangle_{q(\mathbf{x})}\mathbf{C}^\top\mathbf{C} - 2\mathbf{C}^\top\mathbf{V}^{-1}\langle\mathbf{H}\rangle_{q(\mathbf{x})} + (\epsilon\mathbf{J}\mathbf{J}^\top + \boldsymbol{\Omega}^{-1})\mathbf{C}^\top\mathbf{C}\right] + const,$$

The posterior over $\mathbf{C}$ is given by

$$\boldsymbol{\Sigma}_{\mathbf{c}}^{-1} = (\langle\boldsymbol{\Gamma}\rangle_{q(\mathbf{x})} + \epsilon\mathbf{J}\mathbf{J}^\top + \boldsymbol{\Omega}^{-1}) \otimes \mathbf{I}, \tag{57}$$

$$= \boldsymbol{\Sigma}_{\mathbf{C}}^{-1} \otimes \mathbf{I}, \quad \text{where } \boldsymbol{\Sigma}_{\mathbf{C}}^{-1} := \langle\boldsymbol{\Gamma}\rangle_{q(\mathbf{x})} + \epsilon\mathbf{J}\mathbf{J}^\top + \boldsymbol{\Omega}^{-1} \tag{58}$$

$$\boldsymbol{\mu}_{\mathbf{C}} = \mathbf{V}^{-1}\langle\mathbf{H}\rangle_{q(\mathbf{x})}\boldsymbol{\Sigma}_{\mathbf{C}}^\top. \tag{59}$$

Therefore, the approximate posterior over $\mathbf{C}$ is given by

$$q(\mathbf{C}) = \mathcal{MN}(\boldsymbol{\mu}_{\mathbf{C}}, \mathbf{I}, \boldsymbol{\Sigma}_{\mathbf{C}}). \tag{60}$$

The parameters of $q(\mathbf{C})$ depend on the sufficient statistics $\langle\boldsymbol{\Gamma}\rangle_{q(\mathbf{x})}$ and $\langle\mathbf{H}\rangle_{q(\mathbf{x})}$ which are given in section C.1.4.

### C.1.4 Sufficient statistics $\boldsymbol{\Gamma}$ and $\mathbf{H}$

Given the posterior over $\mathbf{x}$, the sufficient statistics $\langle\boldsymbol{\Gamma}\rangle_{q(\mathbf{x})}$ and $\langle\mathbf{H}\rangle_{q(\mathbf{x})}$ necessary to characterise $q(\mathbf{C})$ are computed as follows. Similar to $\langle\mathbf{A}\rangle$, we can simplify $\langle\boldsymbol{\Gamma}_{ij}\rangle_{q(\mathbf{x})}$ as

$$\langle\boldsymbol{\Gamma}_{ij}\rangle_{q(\mathbf{x})} = \gamma^2\sum_{k=1}^{n}\sum_{k'=1}^{n}[\tilde{\mathbf{L}}(k, k') - \tilde{\mathbf{L}}(k, j) - \tilde{\mathbf{L}}(i, k') + \tilde{\mathbf{L}}(i, j)]\,\eta_{ki}\eta_{k'j}\langle\mathbf{x}_k\mathbf{x}_{k'}^\top - \mathbf{x}_k\mathbf{x}_j^\top - \mathbf{x}_i\mathbf{x}_{k'}^\top + \mathbf{x}_i\mathbf{x}_j^\top\rangle_{q(\mathbf{x})}. \tag{61}$$

For $\langle\mathbf{H}_i\rangle_{q(\mathbf{x})}$, we have

$$\langle\mathbf{H}_i\rangle_{q(\mathbf{x})} = \sum_{j=1}^{n}\eta_{ij}\langle(\mathbf{y}_j - \mathbf{y}_i)(\mathbf{x}_j - \mathbf{x}_i)^\top\rangle_{q(\mathbf{x})}, \tag{62}$$

$$= \sum_{j=1}^{n}\eta_{ij}(\mathbf{y}_j\langle\mathbf{x}_j\rangle_{q(\mathbf{x})}^\top - \mathbf{y}_j\langle\mathbf{x}_i\rangle_{q(\mathbf{x})}^\top - \mathbf{y}_i\langle\mathbf{x}_j\rangle_{q(\mathbf{x})}^\top + \mathbf{y}_i\langle\mathbf{x}_i\rangle_{q(\mathbf{x})}^\top), \tag{63}$$

where $\langle\mathbf{x}_i\mathbf{x}_j^\top\rangle_{q(\mathbf{x})} = \boldsymbol{\Sigma}_{\mathbf{x}}^{(ij)} + \langle\mathbf{x}_i\rangle_{q(\mathbf{x})}\langle\mathbf{x}_j\rangle_{q(\mathbf{x})}^\top$ and $\boldsymbol{\Sigma}_{\mathbf{x}}^{(ij)} = \mathrm{cov}(\mathbf{x}_i, \mathbf{x}_j)$.

## C.2 VM step

We set the parameters $\boldsymbol{\theta} = (\alpha, \gamma)$ by maximising the free energy w.r.t. $\boldsymbol{\theta}$:

$$\hat{\boldsymbol{\theta}} = \arg\max_{\boldsymbol{\theta}} \mathbb{E}_{q(\mathbf{x})q(\mathbf{C})}[\log p(\mathbf{y}, \mathbf{C}, \mathbf{x}|\mathbf{G}, \boldsymbol{\theta}) - \log q(\mathbf{x}, \mathbf{C})],$$

$$= \arg\max_{\boldsymbol{\theta}} \mathbb{E}_{q(\mathbf{x})q(\mathbf{C})}[\log p(\mathbf{y}|\mathbf{C}, \mathbf{x}, \mathbf{G}, \boldsymbol{\theta}) + \log p(\mathbf{C}|\mathbf{G}, \boldsymbol{\theta}) + \log p(\mathbf{x}|\mathbf{G}, \boldsymbol{\theta}) - \log q(\mathbf{x}) - \log q(\mathbf{C})]. \tag{64}$$

Once we update all the parameters, we achieve the following lower bound:

$$\mathcal{L}(q(\mathbf{x}, \mathbf{C}), \hat{\boldsymbol{\theta}}) = \mathbb{E}_{q(\mathbf{x})q(\mathbf{C})}[\log p(\mathbf{y}|\mathbf{C}, \mathbf{x}, \mathbf{G}, \hat{\boldsymbol{\theta}})] - D_{KL}(q(\mathbf{C})||p(\mathbf{C}|\mathbf{G})) - D_{KL}(q(\mathbf{x})||p(\mathbf{x}|\mathbf{G}, \hat{\boldsymbol{\theta}})). \tag{65}$$

**Update for $\gamma$**

Recall that the precision matrix in the likelihood term is $\mathbf{V}^{-1} = \gamma \mathbf{I}_{d_y}$. For updating $\gamma$, it is sufficient to consider the log conditional likelihood integrating out $\mathbf{x}, \mathbf{C}$:

$$\mathbb{E}_{q(\mathbf{x})q(\mathbf{C})}[\log p(\mathbf{y}|\mathbf{C}, \mathbf{x}, \mathbf{G}, \boldsymbol{\theta})] = \mathbb{E}_{q(\mathbf{x})q(\mathbf{C})}\left[-\frac{1}{2}\text{Tr}(\boldsymbol{\Gamma}\mathbf{C}^\top\mathbf{C} - 2\mathbf{C}^\top\mathbf{V}^{-1}\mathbf{H}) - \frac{1}{2}\mathbf{y}^\top\Sigma_\mathbf{y}^{-1}\mathbf{y} - \frac{1}{2}\log|2\pi\Sigma_\mathbf{y}|\right], \qquad (66)$$

which is

$$-\frac{1}{2}\mathbb{E}_{q(\mathbf{C})}\text{Tr}(\langle\boldsymbol{\Gamma}\rangle_{q(\mathbf{x})}\mathbf{C}^\top\mathbf{C} - 2\mathbf{C}^\top\mathbf{V}^{-1}\langle\mathbf{H}\rangle_{q(\mathbf{x})}) - \frac{1}{2}\mathbf{y}^\top\Sigma_\mathbf{y}^{-1}\mathbf{y} - \frac{1}{2}\log|2\pi\Sigma_\mathbf{y}|,$$

$$= -\frac{1}{2}\mathbb{E}_{q(\mathbf{C})}[\mathbf{c}^\top(\langle\boldsymbol{\Gamma}\rangle_{q(\mathbf{x})} \otimes \mathbf{I}_{d_y})\mathbf{c} - 2\mathbf{c}^\top\text{vec}(\mathbf{V}^{-1}\langle\mathbf{H}\rangle_{q(\mathbf{x})})] - \frac{1}{2}\mathbf{y}^\top\Sigma_\mathbf{y}^{-1}\mathbf{y} - \frac{1}{2}\log|2\pi\Sigma_\mathbf{y}|,$$

$$= -\frac{d_y}{2}\text{Tr}(\langle\boldsymbol{\Gamma}\rangle_{q(\mathbf{x})}\boldsymbol{\Sigma}_\mathbf{C}) - \frac{1}{2}\text{Tr}(\langle\boldsymbol{\Gamma}\rangle_{q(\mathbf{x})}\boldsymbol{\mu_\mathbf{C}}^\top\boldsymbol{\mu_\mathbf{C}}) + \gamma\text{Tr}(\boldsymbol{\mu_\mathbf{C}}^\top\langle\mathbf{H}\rangle_{q(\mathbf{x})}) - \frac{1}{2}\mathbf{y}^\top\Sigma_\mathbf{y}^{-1}\mathbf{y} - \frac{1}{2}\log|2\pi\Sigma_\mathbf{y}|.$$

The log determinant term is further simplified as

$$-\frac{1}{2}\log|2\pi\Sigma_\mathbf{y}| = -\frac{nd_y}{2}\log(2\pi) + \frac{d_y}{2}\log|\epsilon\mathbf{1}_n\mathbf{1}_n^\top + 2\gamma\mathbf{L}|. \qquad (67)$$

We denote the objective function for updating $\gamma$ by $l(\gamma)$, which consists of all the terms that depend on $\gamma$ above

$$l(\gamma) = -\frac{1}{2}\text{Tr}(\langle\boldsymbol{\Gamma}\rangle_{q(\mathbf{x})}(d_y\boldsymbol{\Sigma}_\mathbf{C} + \boldsymbol{\mu_\mathbf{C}}^\top\boldsymbol{\mu_\mathbf{C}})) + \gamma\text{Tr}(\boldsymbol{\mu_\mathbf{C}}^\top\langle\mathbf{H}\rangle_{q(\mathbf{x})}) - \frac{1}{2}\mathbf{y}^\top((\epsilon\mathbf{1}_n\mathbf{1}_n^\top + 2\gamma\mathbf{L}) \otimes \mathbf{I}_{dy})\mathbf{y} + \frac{d_y}{2}\log|\epsilon\mathbf{1}_n\mathbf{1}_n^\top + 2\gamma\mathbf{L}|,$$

$$= l_1(\gamma) + l_2(\gamma) + l_3(\gamma) + l_4(\gamma),$$

where each term is given below. From the definition of $\boldsymbol{\Gamma} = \mathbf{Q}\tilde{\mathbf{L}}\mathbf{Q}^\top$, we rewrite the first term above as

$$l_1(\gamma) = -\frac{1}{2}\text{Tr}(\langle\mathbf{Q}\tilde{\mathbf{L}}\mathbf{Q}^\top\rangle_{q(\mathbf{x})}(d_y\boldsymbol{\Sigma}_\mathbf{C} + \boldsymbol{\mu_\mathbf{C}}^\top\boldsymbol{\mu_\mathbf{C}})).$$

We separate $\gamma$ from $\mathbf{Q}$ and plug in the definition of $\tilde{\mathbf{L}}$, which gives us

$$l_1(\gamma) = -\frac{1}{2}\gamma^2\text{Tr}(\langle\hat{\mathbf{Q}}\tilde{\mathbf{L}}\hat{\mathbf{Q}}^\top\rangle_{q(\mathbf{x})}(d_y\boldsymbol{\Sigma}_\mathbf{C} + \boldsymbol{\mu_\mathbf{C}}^\top\boldsymbol{\mu_\mathbf{C}})),$$

where the $j$th chunk (of length $d_x$) of $i$th column of $\hat{\mathbf{Q}} \in \mathbb{R}^{nd_x \times n}$ is given by $\hat{\mathbf{q}}_i(j) = \eta_{ij}(\mathbf{x}_i - \mathbf{x}_j) + \delta_{ij}[\sum_k \eta_{ik}(\mathbf{x}_i - \mathbf{x}_k)]$. We can explicitly write down $\tilde{\mathbf{L}}$ in terms of $\gamma$ using orthogonality of singular vectors between $\epsilon\mathbf{1}_n\mathbf{1}_n^\top$ and $2\gamma\mathbf{L}$, where we denote the singular decomposition of $\mathbf{L} = \mathbf{U}_L\mathbf{D}_L\mathbf{V}_L^\top$

$$\tilde{\mathbf{L}} = (\epsilon\mathbf{1}_n\mathbf{1}_n^\top + 2\gamma\mathbf{L})^{-1},$$

$$:= \mathbf{V}_L\begin{bmatrix} 0 & 0 & \cdots & 0 \\ 0 & & \ddots & \vdots \\ 0 & \cdots & \cdots & \frac{1}{\epsilon n} \end{bmatrix}\mathbf{U}_L^\top + \frac{1}{2\gamma}\mathbf{V}_L\begin{bmatrix} \frac{1}{\mathbf{D}_L(1,1)} & 0 & \cdots & 0 \\ 0 & \frac{1}{\mathbf{D}_L(2,2)} & \ddots & \vdots \\ 0 & \cdots & \cdots & 0 \end{bmatrix}\mathbf{U}_L^\top,$$

$$= \tilde{\mathbf{L}}_\epsilon + \frac{1}{2\gamma}\tilde{\mathbf{L}}_L$$

Hence, $l_1(\gamma)$ is given by

$$l_1(\gamma) = -\frac{1}{2}\gamma^2\text{Tr}(\langle\hat{\mathbf{Q}}\tilde{\mathbf{L}}_\epsilon\hat{\mathbf{Q}}^\top\rangle_{q(\mathbf{x})}(d_y\boldsymbol{\Sigma}_\mathbf{C} + \boldsymbol{\mu_\mathbf{C}}^\top\boldsymbol{\mu_\mathbf{C}})) - \frac{\gamma}{4}\text{Tr}(\langle\hat{\mathbf{Q}}\tilde{\mathbf{L}}_L\hat{\mathbf{Q}}^\top\rangle_{q(\mathbf{x})}(d_y\boldsymbol{\Sigma}_\mathbf{C} + \boldsymbol{\mu_\mathbf{C}}^\top\boldsymbol{\mu_\mathbf{C}})).$$

Let $\langle\boldsymbol{\Gamma}_\epsilon\rangle := \langle\hat{\mathbf{Q}}\tilde{\mathbf{L}}_\epsilon\hat{\mathbf{Q}}^\top\rangle_{q(\mathbf{x})}$. Similar to Eq. 61, we have

$$\langle\boldsymbol{\Gamma}_{\epsilon,ij}\rangle = \gamma^2\sum_{k=1}^{n}\sum_{k'=1}^{n}[\tilde{\mathbf{L}}_\epsilon(k,k') - \tilde{\mathbf{L}}_\epsilon(k,j) - \tilde{\mathbf{L}}_\epsilon(i,k') + \tilde{\mathbf{L}}_\epsilon(i,j)]\,\eta_{ki}\eta_{k'j}\langle\mathbf{x}_k\mathbf{x}_{k'}^\top - \mathbf{x}_k\mathbf{x}_j^\top - \mathbf{x}_i\mathbf{x}_{k'}^\top + \mathbf{x}_i\mathbf{x}_j^\top\rangle_{q(\mathbf{x})}.$$

Because $\mathbf{L}$ is symmetric, $\mathbf{U}_L = \mathbf{V}_L$ in the SVD and $\mathbf{U}_L$ contains the eigenvectors of $\mathbf{L}$. So $\tilde{\mathbf{L}}_\epsilon(p,q) = \mathbf{U}_L(p,n)\mathbf{U}_L(n,q)\frac{1}{\epsilon n}$ where we refer to the $n^{th}$ (last) eigenvector of $\mathbf{L}$. However, the last eigenvector of $\mathbf{L}$ corresponding to the eigenvalue 0 has the same element in each coordinate i.e., $\mathbf{U}_L(:,n) = a\mathbf{1}_n$ for some constant $a \in \mathbb{R}$. This implies that

$\tilde{\mathbf{L}}_\epsilon(p, q) = aa\frac{1}{\epsilon n}$. The elements of $\tilde{\mathbf{L}}_\epsilon$ have the same value, implying $[\tilde{\mathbf{L}}_\epsilon(k, k') - \tilde{\mathbf{L}}_\epsilon(k, j) - \tilde{\mathbf{L}}_\epsilon(i, k') + \tilde{\mathbf{L}}_\epsilon(i, j)] = \frac{1}{\epsilon n}[aa - aa - aa + aa] = 0$ and $\langle \mathbf{\Gamma}_{\epsilon,ij} \rangle = 0$ for all $i, j$ blocks. We have

$$l_1(\gamma) = -\frac{\gamma}{4}\text{Tr}(\langle \hat{\mathbf{Q}}\tilde{\mathbf{L}}_L\hat{\mathbf{Q}}^\top \rangle_{q(\mathbf{x})}(d_y\mathbf{\Sigma_C} + \boldsymbol{\mu_C}^\top\boldsymbol{\mu_C})).$$

The second term $l_2(\gamma)$ is given by

$$l_2(\gamma) = \gamma\text{Tr}(\boldsymbol{\mu_C}^\top\langle \mathbf{H} \rangle_{q(\mathbf{x})}),$$

and the third term $l_3(\gamma)$ is rewritten as

$$l_3(\gamma) = -\gamma\text{Tr}(\mathbf{L}\mathbf{Y}^\top\mathbf{Y}).$$

Finally, the last term is simplified as

$$\frac{d_y}{2}\log|\epsilon\mathbf{1}_n\mathbf{1}_n^\top + 2\gamma\mathbf{L}| = \frac{d_y}{2}\left(\sum_{i=1}^{n-1}\log\mathbf{D}_L(i, i) + \log(n\epsilon) + (n-1)\log(2\gamma)\right).$$

Hence,

$$l_4(\gamma) = \frac{d_y}{2}(n-1)\log(2\gamma).$$

The update for $\gamma$ is thus given by

$$\gamma = \arg\max_\gamma l(\gamma) = \arg\max_\gamma l_1(\gamma) + l_2(\gamma) + l_3(\gamma) + l_4(\gamma)$$

$$= -\frac{d_y(n-1)/2}{-\frac{1}{4}\text{Tr}(\langle \hat{\mathbf{Q}}\tilde{\mathbf{L}}_L\hat{\mathbf{Q}}^\top \rangle_{q(\mathbf{x})}(d_y\mathbf{\Sigma_C} + \boldsymbol{\mu_C}^\top\boldsymbol{\mu_C})) + \text{Tr}(\boldsymbol{\mu_C}^\top\langle \mathbf{H} \rangle_{q(\mathbf{x})}) - \text{Tr}(\mathbf{L}\mathbf{Y}^\top\mathbf{Y})}.$$

**Update for $\alpha$**

We update $\alpha$ by maximizing Eq. (64) which is equivalent to maximizing the following expression.

$$-D_{KL}(q(\mathbf{x})||p(\mathbf{x}|\mathbf{G}, \hat{\boldsymbol{\theta}})) = \mathbb{E}_{q(\mathbf{x})q(\mathbf{C})}[\log p(\mathbf{x}|\mathbf{G}, \boldsymbol{\theta}) - \log q(\mathbf{x})] \tag{68}$$

$$= -\int d\mathbf{x}\,\mathcal{N}(\mathbf{x}|\boldsymbol{\mu_x}, \mathbf{\Sigma_x})\log\frac{\mathcal{N}(\mathbf{x}|\boldsymbol{\mu_x}, \mathbf{\Sigma_x})}{\mathcal{N}(\mathbf{x}|\mathbf{0}, \mathbf{\Pi})},$$

$$= \frac{1}{2}\log|\mathbf{\Sigma_x}\mathbf{\Pi}^{-1}| - \frac{1}{2}\text{Tr}\left[\mathbf{\Pi}^{-1}\mathbf{\Sigma_x} - \mathbf{I}_{nd_x}\right] - \frac{1}{2}\boldsymbol{\mu_x}^\top\mathbf{\Pi}^{-1}\boldsymbol{\mu_x}, \tag{69}$$

$$= \frac{1}{2}\log|\mathbf{\Sigma_x}| + \frac{1}{2}\log|\alpha\mathbf{I} + \mathbf{\Omega}^{-1}| - \frac{\alpha}{2}\text{Tr}[\mathbf{\Sigma_x}] - \frac{1}{2}\text{Tr}\left[\mathbf{\Omega}^{-1}\mathbf{\Sigma_x}\right] + \frac{nd_x}{2} - \frac{\alpha}{2}\boldsymbol{\mu_x}^\top\boldsymbol{\mu_x} - \frac{1}{2}\boldsymbol{\mu_x}^\top\mathbf{\Omega}^{-1}\boldsymbol{\mu_x}$$
$$\tag{70}$$

$$:= f_\alpha(\alpha) \tag{71}$$

The stationarity condition of $\alpha$ is given by

$$\frac{\partial}{\partial\alpha}\mathbb{E}_{q(\mathbf{x})q(\mathbf{C})}[\log p(\mathbf{x}|\mathbf{G}, \boldsymbol{\theta}) - \log q(\mathbf{x})] = \frac{1}{2}\text{Tr}((\alpha\mathbf{I} + \mathbf{\Omega}^{-1})^{-1}) - \frac{1}{2}\text{Tr}[\mathbf{\Sigma_x}] - \frac{1}{2}\boldsymbol{\mu_x}^\top\boldsymbol{\mu_x} = 0, \tag{72}$$

which is not closed-form and requires finding the root of the equation.

For updating $\alpha$, we will find $\alpha = \arg\max_\alpha f_\alpha(\alpha)$:

$$\alpha = \arg\max_\alpha \log|\alpha I + \mathbf{\Omega}^{-1}| - \alpha\text{Tr}[\mathbf{\Sigma_x}] - \alpha\boldsymbol{\mu_x}^\top\boldsymbol{\mu_x}. \tag{73}$$

Assume $\mathbf{\Omega}^{-1} = E_\Omega V_\Omega E_\Omega^\top$ by eigen-decomposition and $V_\Omega = \text{diag}(v_{11}, \ldots, v_{nd_x, nd_x})$. The main difficult in optimizing $\alpha$ comes from the first term.

$$\log|\alpha I + \mathbf{\Omega}^{-1}| \overset{(a)}{=} \log|\alpha E_\Omega E_\Omega^\top + E_\Omega V_\Omega E_\Omega^\top| \tag{74}$$

$$= \log|E_\Omega(\alpha I + V_\Omega)E_\Omega^\top| \tag{75}$$

$$\overset{(b)}{=} \log|\alpha I + V_\Omega| = \sum_{i=1}^{nd_x}\log(\alpha + v_{ii}) \tag{76}$$

$$\overset{(c)}{=} d_x\sum_{j=1}^{n}\log(\alpha + 2\omega_i) \tag{77}$$

where at $(a)$ we use the fact that $E_\Omega$ is orthogonal. At $(b)$, the determinant of a product is the product of the determinants, and that the determinant of an orthogonal matrix is 1. Assume that $L = E_L V_L E_L^\top$ by eigen-decomposition and $V_L = \text{diag}\left(\{\omega_i\}_{i=1}^n\right)$. Recall that $\mathbf{\Omega}^{-1} = 2\mathbf{L} \otimes \mathbf{I}_{d_x}$. By Theorem 1, $v_{ii} = 2\omega_i$ and $2\omega_i$ appears $d_x$ times for each $i = 1, \ldots, n$. This explains the $d_x$ factor in $(c)$.

In the implementation, we use fminbnd in Matlab to optimize the negative of Eq. (73) to get an update for $\alpha$. The eigen-decomposition of $L$ (not $\Omega^{-1}$ which is bigger) is needed only once in the beginning. We only need the eigenvalues of $L$, not the eigenvectors.

**KL divergence of C**

$$
\begin{aligned}
& -D_{KL}(q(\mathbf{C})||p(\mathbf{C}|\mathbf{G})) \\
=& \mathbb{E}_{q(\mathbf{x})q(\mathbf{C})}[\log p(\mathbf{C}|\mathbf{G}, \boldsymbol{\theta}) - \log q(\mathbf{c})] \\
=& -\int d\mathbf{c}\, \mathcal{N}(\mathbf{c}|\boldsymbol{\mu_c}, \boldsymbol{\Sigma_c}) \log \frac{\mathcal{N}(\mathbf{c}|\boldsymbol{\mu_c}, \boldsymbol{\Sigma_c})}{\mathcal{N}(0, ((\epsilon\mathbf{JJ}^\top + \mathbf{\Omega}^{-1}) \otimes \mathbf{I})^{-1})}, \\
=& \frac{1}{2}\log|\boldsymbol{\Sigma_c}((\epsilon\mathbf{JJ}^\top + \mathbf{\Omega}^{-1}) \otimes \mathbf{I})| - \frac{1}{2}\text{Tr}\left[\boldsymbol{\Sigma_c}((\epsilon\mathbf{JJ}^\top + \mathbf{\Omega}^{-1}) \otimes \mathbf{I}) - \mathbf{I}\right] - \frac{1}{2}\boldsymbol{\mu_c}^\top((\epsilon\mathbf{JJ}^\top + \mathbf{\Omega}^{-1}) \otimes \mathbf{I})\boldsymbol{\mu_c}, \\
=& \frac{1}{2}\log|(\boldsymbol{\Sigma_C}(\epsilon\mathbf{JJ}^\top + \mathbf{\Omega}^{-1})) \otimes \mathbf{I}| - \frac{1}{2}\text{Tr}\left[(\boldsymbol{\Sigma_C}(\epsilon\mathbf{JJ}^\top + \mathbf{\Omega}^{-1})) \otimes \mathbf{I} - \mathbf{I}\right] - \frac{1}{2}\text{Tr}((\epsilon\mathbf{JJ}^\top + \mathbf{\Omega}^{-1})\boldsymbol{\mu_C}^\top\boldsymbol{\mu_C}), \\
=& \frac{d_y}{2}\log|\boldsymbol{\Sigma_C}(\epsilon\mathbf{JJ}^\top + \mathbf{\Omega}^{-1})| - \frac{d_y}{2}\text{Tr}[\boldsymbol{\Sigma_C}(\epsilon\mathbf{JJ}^\top + \mathbf{\Omega}^{-1})] + \frac{1}{2}nd_xd_y - \frac{1}{2}\text{Tr}((\epsilon\mathbf{JJ}^\top + \mathbf{\Omega}^{-1})\boldsymbol{\mu_C}^\top\boldsymbol{\mu_C}).
\end{aligned}
$$

# D  Connection to GP-LVM

To see how our model is related to GP-LVM, we integrate out $\mathbf{C}$ from the likelihood:

$$
\begin{aligned}
p(\mathbf{y}|\mathbf{x}, \mathbf{G}, \boldsymbol{\theta}) &= \int p(\mathbf{y}|\mathbf{c}, \mathbf{x}, \boldsymbol{\theta})p(\mathbf{c}|\mathbf{G})d\mathbf{c}, \\
&\propto \int \exp\left[-\frac{1}{2}(\mathbf{c}^\top(\mathbf{\Gamma} \otimes \mathbf{I})\mathbf{c} - 2\mathbf{c}^\top\text{vec}(\mathbf{V}^{-1}\mathbf{H})) - \frac{1}{2}\mathbf{y}^\top\Sigma_\mathbf{y}^{-1}\mathbf{y} - \frac{1}{2}\mathbf{c}^\top((\epsilon\mathbf{JJ}^\top + \mathbf{\Omega}^{-1}) \otimes \mathbf{I})\mathbf{c}\right] d\mathbf{c}, \\
&\propto \int \exp\left[-\frac{1}{2}(\mathbf{c}^\top((\mathbf{\Gamma} + \epsilon\mathbf{JJ}^\top + \mathbf{\Omega}^{-1}) \otimes \mathbf{I})\mathbf{c} - 2\mathbf{c}^\top\text{vec}(\mathbf{V}^{-1}\mathbf{H}))\right] d\mathbf{c} - \frac{1}{2}\mathbf{y}^\top\Sigma_\mathbf{y}^{-1}\mathbf{y}, \\
&\propto \exp\left[\frac{1}{2}\text{vec}(\mathbf{V}^{-1}\mathbf{H})^\top((\mathbf{\Gamma} + \epsilon\mathbf{JJ}^\top + \mathbf{\Omega}^{-1}) \otimes \mathbf{I})^{-T}\text{vec}(\mathbf{V}^{-1}\mathbf{H}) - \frac{1}{2}\mathbf{y}^\top\Sigma_\mathbf{y}^{-1}\mathbf{y}\right],
\end{aligned}
$$

where the last line comes from the fact : $\int \exp\left[-\frac{1}{2}\mathbf{c}^\top\mathbf{Mc} + \mathbf{c}^\top\mathbf{m}\right] d\mathbf{c} \propto \exp\left[\frac{1}{2}\mathbf{m}^\top\mathbf{M}^{-T}\mathbf{m}\right]$.

The term $\text{vec}(\mathbf{V}^{-1}\mathbf{H})$ is linear in $\mathbf{y}$ where

$$
\begin{aligned}
\mathbf{H}_i &= \sum_{j=1}^n \eta_{ij}(\mathbf{y}_j - \mathbf{y}_i)(\mathbf{x}_j - \mathbf{x}_i)^\top, \\
&= \sum_{j=1}^n \mathbf{y}_j\eta_{ij}(\mathbf{x}_j - \mathbf{x}_i)^\top - \mathbf{y}_i\sum_{j=1}^n \eta_{ij}(\mathbf{x}_j - \mathbf{x}_i)^\top, \\
&= \tilde{\mathbf{Y}}\mathbf{u}_i + \mathbf{y}_i\mathbf{v}_i,
\end{aligned}
$$

where the vectors $\mathbf{u}_i$ and $\mathbf{v}_i$ are defined by

$$
\begin{aligned}
\tilde{\mathbf{Y}} &= [\mathbf{y}_1 \cdots \mathbf{y}_n], \\
\mathbf{u}_i &= \begin{bmatrix} \eta_{i1}(\mathbf{x}_1 - \mathbf{x}_i)^\top \\ \vdots \\ \eta_{in}(\mathbf{x}_n - \mathbf{x}_i)^\top \end{bmatrix}, \quad \mathbf{v}_i = -\sum_{j=1}^n \eta_{ij}(\mathbf{x}_j - \mathbf{x}_i)^\top.
\end{aligned}
$$

Using these notations, we can write $H$ as

$$
\begin{aligned}
\mathbf{H} &= [\mathbf{H}_1, \cdots, \mathbf{H}_n], \\
&= \tilde{\mathbf{Y}}\mathbf{W},
\end{aligned}
$$

where

$$
\begin{aligned}
\mathbf{W} &= \mathbf{U}_u + \mathbf{V}_v, \\
\mathbf{U}_u &= [\mathbf{u}_1, \cdots, \mathbf{u}_n], \\
\mathbf{V}_v &= \begin{bmatrix}
\mathbf{v}_1 & 0 & \cdots & 0 \\
0 & \mathbf{v}_2 & \cdots & 0 \\
\vdots & 0 & \vdots & 0 \\
0 & \cdots & 0 & \mathbf{v}_n
\end{bmatrix}.
\end{aligned}
$$

So, we can explicitly write down $\mathrm{vec}(\mathbf{V}^{-1}\mathbf{H})$ as

$$
\begin{aligned}
\mathrm{vec}(\mathbf{V}^{-1}\mathbf{H}) &= \mathrm{vec}(\mathbf{V}^{-1}\tilde{\mathbf{Y}}\mathbf{W}), \\
&= (\mathbf{W}^\top \otimes \mathbf{V}^{-1})\mathrm{vec}(\tilde{\mathbf{Y}}), \\
&= (\mathbf{W}^\top \otimes \mathbf{V}^{-1})\mathbf{y}.
\end{aligned}
$$

Using all these, we can rewrite the likelihood as

$$
p(\mathbf{y}|\mathbf{x}, \mathbf{G}, \boldsymbol{\theta}) \quad \propto \quad \exp\left[ -\frac{1}{2}\mathbf{y}^\top \mathbf{K}_{LL}^{-1}\, \mathbf{y} \right],
$$

where the precision matrix is given by

$$
\begin{aligned}
\mathbf{K}_{LL}^{-1} &= \boldsymbol{\Sigma}_{\mathbf{y}}^{-1} - (\mathbf{W}^\top \otimes \mathbf{V}^{-1})^\top \boldsymbol{\Lambda}(\mathbf{W}^\top \otimes \mathbf{V}^{-1}), \\
\boldsymbol{\Lambda} &= ((\boldsymbol{\Gamma} + \epsilon\mathbf{J}\mathbf{J}^\top + \boldsymbol{\Omega}^{-1}) \otimes \mathbf{I})^{-T}.
\end{aligned}
$$

# E    Useful results

In this section, we summarize theorems and matrix identities useful for deriving update equations of LL-LVM. The notation in this section is independent of the rest.

**Theorem 1.** *Let $A \in \mathbb{R}^{n \times n}$ have eigenvalues $\lambda_i$, and let $B \in \mathbb{R}^{m \times m}$ have eigenvalues $\mu_j$. Then the $mn$ eigenvlaues of $A \otimes B$ are*

$$
\lambda_1\mu_1, \ldots, \lambda_1\mu_m, \lambda_2\mu_1, \ldots, \lambda_2\mu_m, \ldots, \lambda_n\mu_m.
$$

**Theorem 2.** *A graph Laplacian $L \in \mathbb{R}^{n \times n}$ is positive semi-definite. That is, its eigenvalues are non-negative.*

## E.1    Matrix identities

$$
x^\top(A \circ B)y = \mathrm{tr}(\mathrm{diag}(x)A\,\mathrm{diag}(y)B^\top) \tag{78}
$$

From section 8.1.1 of the matrix cookbook [1],

$$
\int \exp\left[ -\frac{1}{2}x^\top Ax + c^\top x \right]\, \mathrm{dx} = \sqrt{\det(2\pi A^{-1})} \exp\left[ \frac{1}{2}c^\top A^{-\top}c \right]. \tag{79}
$$

**Lemma 1.** *If $X = (x_1|\cdots|x_n)$ and $C = (c_1|\ldots|c_n)$, then*

$$
\int \exp\left[ -\frac{1}{2}\mathrm{tr}(X^\top AX) + \mathrm{tr}(C^\top X) \right]\, \mathrm{d}X = \det(2\pi A^{-1})^{n/2}\exp\left[ \frac{1}{2}\mathrm{tr}(C^\top A^{-1}C) \right].
$$

Woodbury matrix identity

$$
(A + UCV)^{-1} = A^{-1} - A^{-1}U(C^{-1} + VA^{-1}U)^{-1}VA^{-1}. \tag{80}
$$

## Footnotes

[1] The equivalent expression in term of matrix normal distribution for $\mathbf{Y} = [\mathbf{y}_1, \mathbf{y}_2, \cdots, \mathbf{y}_n] \in \mathbb{R}^{d_y \times n}$

# References

[1] K. B. Petersen and M. S. Pedersen. The matrix cookbook, nov 2012. Version 20121115.