[Reviews · NeurIPS 2015]

Submitted by Assigned_Reviewer_1

The paper introduces a model which is probabilistic for non linear manifold discovery. It is based on a generative model with missing variables and required a variational EM implementation which is standard but nevertheless technical to derive in this specific context.

As regards the bibliography, I can see at least two relevant references missing: [1] A. Deleforge, F. Forbes and R. Horaud, High-Dimensional Regression with Gaussian Mixtures and Partially-Latent Response Variables. Statistics and Computing.

[2] A. Deleforge, F. Forbes, S. Ba and R. Horaud, Hyper-spectral image analysis with Partially-Latent Regression and spatial Markov dependencies. IEEE journal of selected topics in signal processing 2015.

Note that [2] is an extension of [1] with a Markov neighborhood and a VEM approach to high-to-low dimensional mapping.

In terms of results, the authors seem to stop their EM algorithm after a rather small (30-40) number of iterations without any mention of any more formal convergence or stoppping criterion. Why is it so? This looks a little suspicious. Do the results degrade when more iterations are done?

In figure 3 B, I cannot see the mentioned graph structure?

Comparison with GP-LVM: this paragraph is missing a clear conclusion at the end. I'd suggest the authors explain better what they conclude from this comparison.

Fig 5 F: is the difference between the first 3 methods actually significant? Fig 5-A: The authors mention multiple initializations of EM but what is their conclusion wrt the sensitivity of their algorithm to initialization?

Fig 6: I would need more information as why the authors think that the result of their method, although better than the others, is satisfying. It seems to me, at least visually, that the recovered mapping is quite far from the ground truth one.

In their conclusion, the authors mention that their model permits evaluation of hypothetical neighborhood but I do not see where in the paper this is illustrated.

Summary: The paper is well written and proposes an interesting model for non linear manifold discovery based on a generative approach. The results section is not fully convincing but is ok.

Submitted by Assigned_Reviewer_2

This paper proposes a Bayesian probabilistic model for manifold learning. The model captures a manifold as in locally linear spaces.

The main weakness of this paper is that the novelties of the proposed method are insufficient, unclear, and/or unconvincing. In my understanding, the advantages of the method described in section 1 is summarized as follows: The model

(1) is a Bayesian probabilistic model and thus can handle uncertainty (2) can do model selection (3) does not need some extra step that other method need

(1) is just a common nature of probabilistic models. (2) is good but other methods can do the same thing with e.g. cross validation. For (3), I couldn't find any explanations about the extra step so I couldn't judge this advantage.

Summary: The novelties of the proposed method are insufficient, unclear, and/or unconvincing.

Author Feedback
Author rebuttal: We thank the reviewers for their positive feedback on our paper.

Rev 2 and 5 asked the difference between LL-LVM and GP-LVM: As noted in lines 65-68 in introduction, both are Gaussian models, but GP-LVM is parameterised by a stationary covariance function rather than by a local neighbour-defined precision matrix which seeks to explicitly preserve local neighbourhood properties.

Rev 1

We will add the complexity analysis. An expensive step in the E-step is the inversion of the posterior precision matrices of both x and C (in eq. 41 and 55 in the Appendix) which costs roughly cubic in # observations with naive implementation. We will study possible ways to decrease the complexity in future work.

Rev 2

Missing references: Thanks for pointing them out. With x denoting a latent variable and y denoting an observation, the suggested works (GLLiM) consider p(y|x, z) to be a Gaussian with mean A_z x+b_z where A is to be learned and z is a latent discrete random variable. Although its formulation and probabilistic nature certainly are similar to our approach, GLLiM does not have the goal of preserving the local neighbourhood structure of the data manifold. We will address this in the final version.

Stopping EM: Although we did not use a formal stopping criterion, we noted the usual behaviour of EM-like ascent of saturation in the free-energy (see Fig 5A). Running further iterations did not change the structure of the solution, although the scaling of x and C (which is degenerate in the likelihood) could change slightly leading to very minor gains in free-energy.

Graph in Fig 3B: The graphical structure was represented with grey lines for graph edges, indicating neighbouring data points.

Significance of classification errors in Fig. 5F: While we conducted only 5-fold cross-validation and so statistical power is weak, we observed that the GP-LVM classification error was higher than that of LL-LVM on all 5 splits, and higher than that of ISOMAP on 4 (equal on the last). ISOMAP error was smaller than LL-LVM in 3 cases, larger in 1 and equal in the other. Formally, using 2-sided tests, the hypothesis of equal performance could be rejected for the GP-LVM - LL-LVM (p = 0.003) and GP-LVM - ISOMAP (p = 0.02) comparisons by a t-test. Without the normality assumption (i.e. sign test), we had insufficient power to reject any hypothesis of equality (the smallest possible p-value with 5 paired points is 0.0625, achieved for LL-LVM over GP-LVM). Thus we believe our statement that 'Classification with LL-LVM coordinates outperforms GP-LVM and LLE, and matches ISOMAP' is not unreasonable.

Sensitivity to initialization: The free energy is indeed multimodal, as is common in many latent variable models (including GP-LVM). Restarting EM from different initial conditions and choosing the solution with largest free energy is a common response to this problem.

Climate data: We would like to emphasize that each weather station is represented as a 12-dimensional vector of monthly precipitation measurements. Given only such information, recovering the exact 2D physical coordinates of the stations is an extremely challenging task and not our main purpose. Our aim is to recover 2D geographical relationships between weather stations i.e., physically nearby stations are also close in the recovered low-dimensional space.

Hypothetical neighbourhood: We demonstrated the ability to evaluate specific hypotheses about short-cuts in Sec. 4.1 and Fig 4, and different neighbourhood sizes in Fig 5A.

Rev 3

We agree with the reviewer's first two general comments. It is important to note, however, that LL-LVM is the only probabilistic approach for manifold recovery we are aware of that is built on the neighbourhood structure of the data. And cross-validation requires both a cost function and the ability to evaluate that function on held out data. Out-of-sample extension, and thus validation, has proven challenging for non-probabilistic methods. We would be grateful if the reviewer can give concrete examples of related manifold-based methods with model selection capability.

Regarding "the extra step", we referred to refs 9-11 which learn multiple local models yielding multiple different local coordinates. An extra alignment step is then needed to "stitch" the local coordinates together to get one global coordinate system. In contrast, LL-LVM's objective allows it to learn globally consistent coordinates naturally, thus no extra step is needed.

Rev 6

Yes, it is more flexible to consider full covariance to capture the different scales and different correlation structure along both directions of C. However, in that case, the size of the full covariance matrix is prohibitively large (n*dy*dx by n*dy*dx) , while the current form is (n*dx) by (n*dx), given that dy is typically 100s to 10,000s. Having the scaled covariance can save computations and storage requirements significantly.